# Can changes in deformation regimes be inferred from crystallographic preferred orientations in polar ice?

Maria-Gema Llorens[1*], Albert Griera[2], Paul D. Bons[3,4], Ilka Weikusat[3,5], David J. Prior[6], Enrique Gomez-Rivas[7], Tamara de Riese[3], Ivone Jimenez-Munt[1], Daniel García-Castellanos[1] and Ricardo A. Lebensohn[8]

[1] Geosciencies Barcelona CSIC, Lluis Sole i Sabaris s/n, 08028 Barcelona, Spain

[2] Departament de Geologia, Universitat Autònoma de Barcelona, 08193 Cerdanyola del Vallès, Barcelona, Spain

[3] Department of Geosciences, Eberhard Karls University Tübingen, Wilhemstr. 56, 72074 Tübingen, Germany

[4] China University of Geosciences, Beijing, China

[5] Alfred Wegener Institute Helmholtz Centre for Polar and Marine Research, Germany

[6] Department of Geology, University of Otago, 362 Leith Street, Dunedin 9016. New Zealand

[7] Departament de Mineralogia, Petrologia i Geologia Aplicada, Facultat de Ciències de la Terra, Universitat de Barcelona, Martí i Franquès s/n, 08028 Barcelona, Spain

[8] Theoretical Division. Los Alamos National Laboratory. Los Alamos, NM, 87545, USA

[*]*Corresponding author:* mgllorens@geo3bcn.csic.es (Maria-Gema Llorens)

***Keywords:*** *ice microstructures, crystallographic preferred orientation, ice-sheet flow, finite strain, flow transitions*

## Abstract

Creep due to ice flow is generally thought to be the main cause for the formation of crystallographic preferred orientations (CPOs) in polycrystalline anisotropic ice. However, linking the development of CPOs to the ice flow history requires a proper understanding of the ice aggregate's microstructural response to flow transitions. In this contribution the influence of ice deformation history on the CPO development is investigated by means of full-field numerical simulations at the microscale. We simulate the CPO evolution of polycrystalline ice under combinations of two consecutive deformation events up to high strain, using the code VPFFT/ELLE. A volume of ice is first deformed under co-axial

boundary conditions, which results in a CPO. The sample is then subjected to different

boundary conditions (co-axial or non-coaxial) in order to observe how the deformation regime switch impacts the CPO. The model results indicate that the second flow event tends to destroy the first, inherited fabric, with a range of transitional fabrics. However, the transition is slow when crystallographic axes are critically oriented with respect to the second imposed regime. Therefore, interpretations of past deformation events from observed CPOs

must be carried out with caution, particularly, in areas with complex deformation histories.

## 1. Introduction

During the last two decades sea level rise has accelerated in association with global climate

change (Nerem et al., 2018), but the limited knowledge available on how fast ice flows in ice sheets along with uncertain boundary conditions (Edwards et al., 2021) give a wide range in long-term sea level rise predictions. The rheological changes of polar ice during its deformation are crucial for the accurate projection of ice sheet discharge into the ocean (Golledge et al., 2015). Polycrystalline ice (ice *1h*) in glaciers, ice sheets and ice shelves

flows in response to gravitational forces (e.g., Cuffey and Paterson, 2010). Crystal-plastic strain of ice is mainly accommodated by the glide of dislocations along their individual crystallographic slip systems (Duval, 1983). Deformation of a grain aggregate by dislocation creep leads to a crystallographic preferred orientation (CPO), also called fabric or texture. The CPO evolves according to the flow kinematics, magnitude of strain and temperature

(Budd and Jacka, 1989; Katayama and Karato, 2006; Qi et al., 2019; Fan et al., 2020), and importantly depends on the activity of, or ease of glide of dislocations on the different slips systems. An ice crystal is strongly anisotropic because it deforms predominantly by the glide of dislocations on the basal plane (0001) and along the *a*-axes <11-20> direction (Duval et al., 1983). The gravity-driven flow of ice sheets produces rotations of ice crystal lattices,

developing CPOs according to the stress configuration (Treverrow et al., 2012). The *c*-axes (normal to the basal plane) tend to get oriented in a cone-shaped distribution of *c*-axes with the cone axis parallel to the maximum compressional finite strain (Qi et al., 2017). *a*-axes tend to be parallel to the maximum extensional finite strain at low temperature and high strain (Qi et al., 2019; Journaux et al., 2019), while they present a tendency to become parallel to

the vorticity axis at high temperature (Qi et al., 2019), also according to CPO descriptions of ice natural samples (Monz et al., 2021; Thomas et al., 2021) For that reason, the observed CPO in ice cores drilled in ice sheets and glaciers is assumed as a reliable macroscopic flow

indicator (Hudleston, 1977; Alley, 1988). Microstructures described from ice cores drilled in Antarctica and Greenland have allowed the interpretation of deformation conditions (see table 1 in Faria et al., 2014). However, because ice can be affected by differences in temperature and stress configurations during ice-sheet flow, unravelling the ice deformation history from $c$-axis fabrics observed in natural ice samples can be challenging. The fact that most deep ice cores are drilled at ice-sheet divides or domes, with the aim of providing the best-quality paleoclimate record, does not help in gaining understanding of natural ice flow from the recovered ice samples.

Most of the current knowledge of the link between ice deformation and CPO development is derived from laboratory experiments. Experimental studies have utilised ice to understand how CPOs develop and evolve under deformation (Kamb et al., 1972; Wilson, 1982; Jacka and Maccagnan, 1984, Wilson and Peternell, 2012; Budd et al., 2013; Montagnat et al., 2015; Vaughan et al., 2017, Fan et al., 2020). Most studies by far use bulk isotropic ice (i.e., with a random CPO) that is then subjected to a single deformation event. Due to the limitations of laboratory deformation experiments, to our knowledge only a few studies have used polar ice samples starting with a pre-existing CPO. Moreover, most of such studies focused on vertical uniaxial compression of samples with a pre-existing CPO that was formed by vertical compression (resulting in a $c$-axis cluster or cone) (Azuma and Higashi, 1984; Dahl-Jensen et al., 1997, Castelnau et al., 1998, Jacka and Li, 2000, Fan et al., 2020). Exceptions are the experiments by Jun and Jacka (1998) and Treverrow et al. (2012), in which the compressional CPO is deformed under horizontal shear, and Craw et al. (2018), where vertical axial compression is applied to sub-samples taken at different angles from a sample with a pre-existing vertical girdle orientation. Numerical modelling provides an alternative approach to overcome these limitations because it allows a continuous analysis of multiple deformation scenarios with a wide variety of deformation kinematics and environmental parameters. Moreover, numerical simulations of polycrystalline ice and their comparison with experimental and natural data provide useful insights into the study of the CPO development, as they allow visualizing and quantifying the microstructural evolution up to high strain (Montagnat et al., 2014b; Piazolo et al., 2019). However, as in the case of laboratory experiments, most numerical studies to date have focused on systems starting with an initially random CPO to which a single deformation event is applied (Montagnat et al., 2011; Llorens et al., 2017, 2020). Although very useful, these experimental and numerical studies can only represent a limited range of real natural scenarios, where ice aggregates with no strain history

are subjected to stress. To date only a few modelling studies have addressed the issue of the evolution of polycrystalline ice that has previously experienced flow. Examples are Thorsteinsson et al. (2003), which evaluates to what extent an observed CPO tracks the local stress within an ice sheet, and Jansen et al. (2016), where the viscoplastic response of ice polycrystals with a starting CPO is studied. More recently, Lilien et al. (2021) analyse whether the effect of changes in ice-stream flow can be recorded in ice-crystal fabrics.

Considering that volumes of polar ice typically experience multiple changes in the deformation regime along their flow path during ice-sheet flow, such as the transition from the co-axial strain in the centre of the ice mass to non-coaxial strain at depth and away from the centre (Jennings and Hambrey, 2021), systematic studies providing a comprehensive understanding of CPO development during multi-stage deformation histories are essential. Hence, this contribution intends to fill this knowledge gap by providing a study of the multi-stage deformation of ice samples and the resulting CPOs for different settings. We present a series of numerical simulations of polycrystalline ice that is affected by two consecutive deformation events. We choose combinations of subsequent deformation kinematics that may be expected to occur in ice sheets. We analyse the CPO developed during these deformation events in order to determine the kinematic conditions for the preservation, modification or destruction of CPOs in ice.

## 2. Flow regime transitions in polar ice sheets

Ice flows in all directions from the accumulation zone towards the edges of the ice mass driven by the gravity force. Due to precipitation and accumulation, ice is gradually compressed ranging from vertical axial compression in domes, to plane strain vertical compression in ridges (zone I in Fig. 1a). Observations from natural ice core samples at these domains range from a vertical single maximum to a vertical girdle (Montagnat et al., 2012; Weikusat et al., 2017). At depth and away from the centre of the ice sheet (zone II in Fig. 1a), ice experiences a vertical gradient in velocity, resulting in simple shear deformation parallel to the bedrock (Hudleston, 2015). At depth, microstructural descriptions from ice cores performed in domes and ridges indicate a vertical single maximum (Thorteinsson et al., 1997; Azuma et al., 1999; Durand et al., 2007; Faria et al., 2014; Weikusat et al., 2017). Inside a glacier, ice stream or in a flank flow (Voigt, 2017) flow acceleration may dominate, resulting in extension along the flow direction (zone III in Fig. 1a), observed by a vertical girdle in ice

core samples (Voigt, 2017). In a shear margin of glaciers and ice streams, as well as in some ice shelves, ice experiences a gradient in velocity in the lateral direction perpendicular to that velocity, resulting in simple shear deformation with a vertical shear plane (zone III and IVa in Fig.1a-b), observed by a single or double maxima (Jackson and Kamb, 1997) oriented sub

perpendicular to the shear plane (Monz et al., 2021; Thomas et al., 2021) and consistent with geophysical data (Smith et al., 217; Lutz et al., 2020, 2022).

In this contribution, we analyse different examples of flow changes that represent relevant and/or common deformation regimes in ice sheets (zones I to IV in Figure 1), assuming a

constant strain rate and temperature.

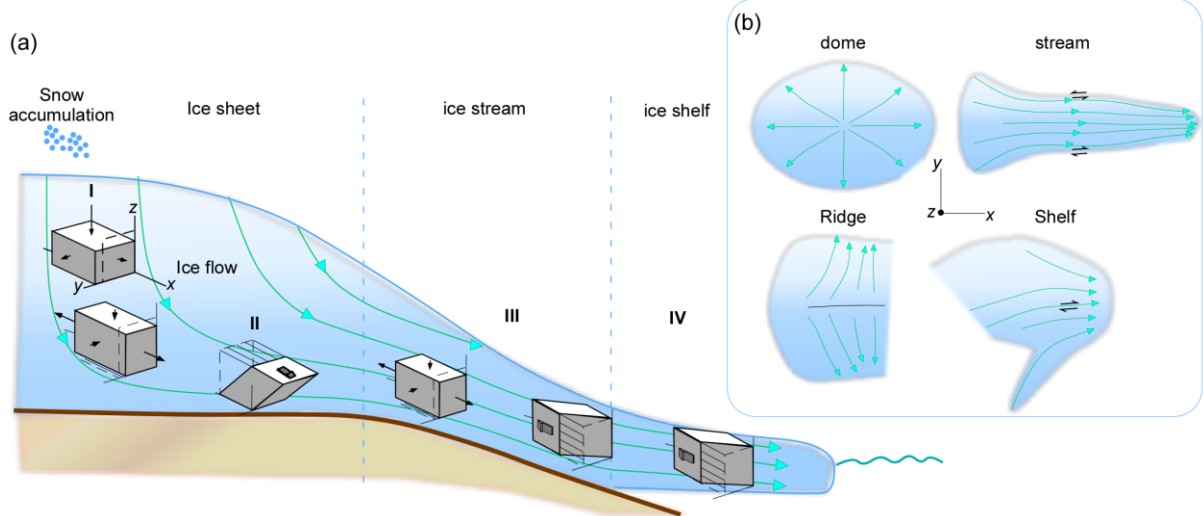

**Figure 1.** Sketch of ice-sheet flow patterns in (a) cross section and (b) map view, showing some dominant deformation conditions at different zones of the ice mass. At the upper and central parts of the dome the dominant flow regime is vertical axial compression parallel to $z$ (indicated by zone I). In ice ridges, an extension

flow transverse to the ridge can occur (Wang et al., 2002). Flow turns progressively more non-coaxial with depth and distance from the dome, where strongly non-coaxial deformation dominates close to the bedrock (indicated by zone II). The dominant kinematic regime at the ice stream is extension along the flow direction, where simple shear deformation with a vertical shear plane can occur in shear margins of the stream (indicated by zone III). At the ice shelf, the predominant regime assumed in this contribution is simple shear in the

horizontal plane ($xy$) (zone IV) (Lutz et al., 2020). The $xyz$ reference frame is defined with $z$ vertical, $x$ horizontal and parallel to flow and $y$ normal to $xz$:

## 3. Methods

### 3.1. Numerical simulation of ice fabrics and postprocessing

Ice polycrystalline viscoplastic deformation was simulated using the Fast Fourier Transform algorithm (VPFFT; Lebensohn and Rollett, 2020), within the numerical open-source platform ELLE (http://www.elle.ws; Bons et al., 2008). The coupling of the full-field VPFTT code and ELLE allows simulating deformation of a polycrystalline aggregate by dislocation glide up to high strains (>10) as often found in nature in ice and rocks. The reader is referred to Lebensohn and Rollett (2020) for a detailed description of the VPFFT approach, and to Griera et al. (2013) and Llorens et al. (2016a) for a detailed description of the coupling between VPFFT and ELLE. The local mechanical response of a nonlinear heterogeneous material can be calculated as a convolution integral of the Green functions associated with a linear homogenous medium and a polarisation field. The VPFFT formulation is used to transform the polarisation field that contains all the information on the heterogeneity and non-linearity of the material's behaviour into Fourier space. By the conversion of the real space convolution integrals to simple products in the Fourier space, the mechanical fields are calculated, and the convolution product is transformed back to real space. In this full-field approach, the semi two-dimensional data structure is discretised by a regular mesh of 256x256 unconnected nodes (*unodes*) or Fourier points (Fig. 2b). The strain rate and stress field under compatibility and equilibrium constraints related by the constitutive equation (1) is obtained by iteratively solving for the flow law for every *unode* (*u*):

$$\dot{\varepsilon}_{ij}(u) = \sum_{s=1}^{N} m_{ij}^s(u)\dot{\gamma}^s(u) = \dot{\gamma}_0 \sum_{s=1}^{N} m_{ij}^s(u) \left(\frac{\left|m_{ij}^s(u)\sigma_{ij}'(u)\right|}{\tau^s(u)}\right)^n \times \text{sgn}\left(m_{ij}^s(u)\sigma_{ij}'(u)\right)$$

(1)

Where $\dot{\varepsilon}_{ij}$ is the strain rate, $\sigma_{ij}'$ is the deviatoric stress, $m_{ij}^s$ is the symmetric Schmid tensor, $\dot{\gamma}^s$ the shear strain rate, $\tau^s$ is the critical resolved shear stress defined for the slip system (*s*), $\dot{\gamma}_0$ is a reference strain rate and *n* is the stress exponent.

In all three slip systems, the shear strain rate is assumed to be related to the deviatoric stress by a stress exponent of *n*=3, in accordance with the "Glen's law" (Haefeli,1961). After convergence, the VPFFT code calculates the associated lattice rotations from the velocity gradient and stress fields. The actual stress exponent in ice 1h may actually be closer to *n*=4 (Bons et al., 2018; Qi and Goldsby, 2021). The first tests show that raising *n* to 4 has little effect on the CPO, and in order to be consistent with previous ice CPO simulations (Llorens

et al., 2016a, 2016b, Steinbach et al., 2016, Jansen et al., 2016; Qi et al., 2019; Piazolo et al., 2019) all simulations presented here are carried out with the more commonly used $n$=3. Uniaxial shortening of a single crystal by glide only along the non-basal planes requires about 60 times higher stress than when the crystal can deform entirely by basal-plane glide (Duval et al., 1983). We therefore set $\tau^s$ 60 times lower for basal glide than for prismatic and pyramidal glide.

Previous numerical studies have shown that about the same single-maximum CPO develops at low temperature and a high strain rate conditions, regardless of the set rate of dynamic recrystallisation (see supplementary figure 4 or Llorens et al., 2016a; 2016b). As this contribution aims to study the CPO response to a change in the deformation regime, the numerical procedures that simulate dynamic recrystallisation (Llorens et al., 2017) are deliberately not incorporated. Every *unode* represents a *crystallite* where the crystal orientation, strain rate, dislocation density and local stress is stored (Fig.2b). All *unodes* inside a grain have the same initial crystal orientation (Fig.2a). The crystal orientation is defined by the three Euler angles. We use the three Euler angles stored at every *unode* of the model for the calculation of the orientation density function (ODF) that provides the orientation densities in Euler space, using the open-source texture analysis software MTEX (Bachmann et al., 2011). The intensity of the CPO is shown by the misorientation index (M-index; Skemer et al., 2005) calculated from the ODF. Crystal symmetry shows the relative proportion of point (P), girdle (G) and random (R) components of the (0001) crystallographic axis, or *c*-axis distribution in a ternary diagram (Vollmer, 1990). The P, G and R proportion is calculated from the three eigenvalues ($a_1$, $a_2$, $a_3$) as P=$a_1 - a_2$, G=$2a_2 - a_3$ and R=$3a_3$.

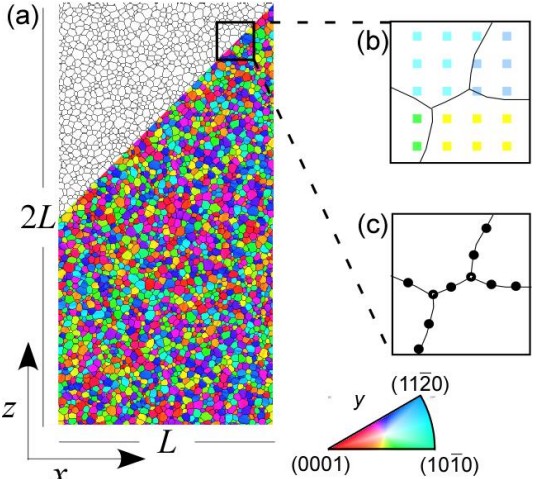

**Figure 2**. (a) The initial microstructure is a 2 x 1 model unit length (*L*). Two data layers are

used in the ELLE approach: (b) the microstructure is discretised by a mesh of 256x256 *unodes* or Fourier points. (c)The boundary nodes (*bnodes*) layer defines grains.

### 3.2. Simulated flow regimes and applied boundary conditions

Deformation was applied to the microstructure by incremental steps of a shear strain of $\gamma$=0.04 or, alternatively, 2% of shortening or extension (Table 1), both equivalent to 0.02 natural strain, defined as $\ln(L_f/L_i)$ (with $L_f$ and $L_i$ being the final and initial length of a line in the direction of maximum extension, respectively). Each example considered in this contribution consists of two deformation regimes that are applied in succession (see Table 2 and supplementary figure 5). We considered four different model series (series A to D) to simulate flow transitions between pairs of deformation regimes (**V**) that dominate in different zones of the ice mass through which a volume of ice may travel.

Series A represents ice flowing from the centre of a dome to deep lateral zones (from zone I to zone II in Fig. 1). To simulate this transition, we carried out a series of simulations with first vertical uniaxial compression parallel to $z$ (**V₁**), followed by top-to-the-right simple shear in the vertical plane (*xz*) with a horizontal shear plane (**V₂**) (Tables 1 and 2). Similar to A, Series B shows the transition of ice flowing in central parts of the ice sheet, but in this case from the centre of a ridge to deep lateral zones (from zone I to zone II in Fig. 1). For series B, we considered that the ice aggregate is first deformed by **V₃**, horizontal uniaxial extension parallel to $x$, followed by **V₂**, dextral simple shear in the vertical plane (*xz*) as in Series A (Tables 1 and 2). Series C simulates ice flowing from an ice dome to an ice flank or stream (from zone I to zone III in Figure 1). Series C was carried out assuming first vertical uniaxial compression parallel to $z$ (**V₁**) followed by uniaxial extension parallel to $x$ (**V₃**) (Table 1 and 2). Finally, series D represents ice flowing from an ice-stream or glacier to an ice shelf or shear margin (from zone III to zone IV in Figure 1). For this series, we considered first uniaxial extension in the $x$ direction (**V₃**), and subsequent dextral simple shear in the horizontal plane (*xy*) with a vertical shear plane parallel to the flow direction (**V₄**) (Tables 1 and 2). As a ridge can be represented by confined compression and extension, rather than pure extension, a sensitivity test of series B, C and D assuming pure shear (**V'₃**) (Table 1) can be found in supplementary figures 1, 2 and 3. For comparison, results from simulations of microstructures deformed under a single-deformation event (**V₂**, **V₃** and **V₄**) are shown together with all series results.

**Table 1**. Velocity gradient tensors **V** considered for the different regimes and corresponding zones in Figure 1. The *xyz* reference frame is defined with *z* vertical, *x* horizontal and parallel to flow and *y* normal to *xz*.

| Name | Regime | Zone (Figure 1) | Velocity gradient tensor |
|------|--------|-----------------|--------------------------|
| $\mathbf{V_1}$ | Vertical uniaxial compression parallel to *z*, extension in *x* and *y* | I | $\begin{bmatrix} 0.01 & 0 & 0 \\ 0 & -0.02 & 0 \\ 0 & 0 & 0.01 \end{bmatrix}$ |
| $\mathbf{V_2}$ | Dextral simple shear in the vertical plane (*xz*), plane strain | II | $\begin{bmatrix} 0 & 0.04 & 0 \\ 0 & 0 & 0 \\ 0 & 0 & 0 \end{bmatrix}$ |
| $\mathbf{V_3}$ | Horizontal uniaxial extension parallel to *x*, compression in *y* and *z* | I, III | $\begin{bmatrix} 0.02 & 0 & 0 \\ 0 & -0.01 & 0 \\ 0 & 0 & -0.01 \end{bmatrix}$ |
| $\mathbf{V'_3}$ | Pure shear, plane strain | I, III | $\begin{bmatrix} 0.02 & 0 & 0 \\ 0 & 0.02 & 0 \\ 0 & 0 & 0 \end{bmatrix}$ |
| $\mathbf{V_4}$ | Dextral simple shear in the horizontal plane (*xy*), plane strain | III, IV | $\begin{bmatrix} 0 & 0 & -0.04 \\ 0 & 0 & 0 \\ 0 & 0 & 0 \end{bmatrix}$ |

**Table 2**. Deformation regimes applied to the different series of numerical simulations, including idealised deformation regimes in drill cores and examples. References of ice core description: GRIP (Thorsteinsson et al., 1997), GISP2 (Gow et al., 1997), Dome C (Durand et al., 2007), Dome F (Azuma et al., 1999), Talos Dome (Montagnat et al., 2012), NorthGRIP (Faria et al., 2014); EDML (Weikusat et al., 2017), NEEM (Montagnat et al., 2014a), Vostok (Lipenkov et al., 1998)

| Name | First + Second regime | Application of the second regime | Ice core | Drill core regime |
|------|-----------------------|----------------------------------|----------|-------------------|
| Series A | $\mathbf{V_1} + \mathbf{V_2}$ | At the last step ($\varepsilon_l$=0.92) | GRIP, GISP2, Dome C, Dome F, Talos Dome | Dome/Summit |
| Series B | $\mathbf{V_3} + \mathbf{V_2}$ | At the last step ($\varepsilon_l$=0.92) | NorthGRIP, EDML, NEEM | Divide |
| Series C | $\mathbf{V_1} + \mathbf{V_3}$ | At steps of $\varepsilon_l$ = 0, 0.2, 0.4, 0.8 and 1.2 | Spice, Vostok | Flank flow |

| | | | | |
|---|---|---|---|---|
| Series D | $\mathbf{V_3 + V_4}$ | At the last step ($\varepsilon_I$=0.92) | | Shear margin |

## 4. CPO evolution results

### 4.1. Series A: Ice flowing from the centre of the dome to deep lateral zones

Snow precipitation at a dome causes vertical uniaxial compression and flattening of ice layers, and outflow in all horizontal directions (zone I in Fig. 1). As ice flows away from domes and gets buried deeply it enters the zone dominated by simple shear parallel to the bedrock (zone II in Fig. 1). To simulate this transition, the initial randomly oriented microstructure in series A is first subjected to vertical uniaxial compression parallel to $z$ ($\mathbf{V_1}$), followed by dextral simple shear in the vertical plane ($xz$) ($\mathbf{V_2}$) (Tables 1 and 2). The pole figures in Figure 3 show the crystallographic preferred orientation evolution during deformation in Series A. The initially untextured microstructure (*i.e.*, with an initially homogeneous and random lattice orientation, see Fig. 3a) develops a strong CPO due to the rotation of $c$-axes towards the maximum shortening direction, according to the imposed vertical compression (see Fig.3a at $\varepsilon_I$=0.92). The observed CPO symmetry, expressed as the proportion of point (P), girdle (G) and random (R) components of $c$-axes (0001), is characterised by a point maximum with no girdle component (Fig. 3c). The final step of the first regime ($\varepsilon_I$=0.92) is used as the initial texture for the second deformation regime simulation, where the aggregate is subjected to simple shear in the vertical plane ($xz$) up to a total natural strain of 4 (see $\varepsilon_2$=3 Fig. 3b).

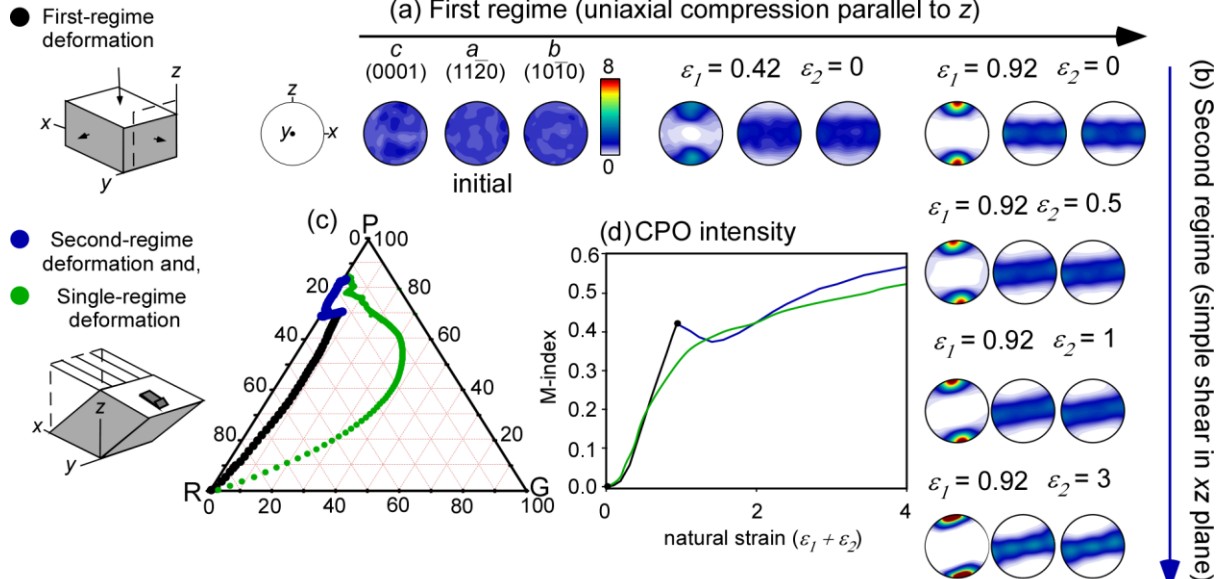

**Figure 3.** CPO evolution during the two consecutive deformation regimes of series A: (a) vertical uniaxial compression parallel to *z* (black) and subsequently (b) dextral simple shear in the vertical plane *xz* (blue). The final texture developed at the end of the first flow regime is used as the initial texture (input) for simulating the second regime. (c) Ternary diagram that shows the CPO symmetry expressed as the proportion of point (P), girdle (G) and random (R) components of *c*-axes (0001). Points plotted at the PGR graph represent every step of deformation ($\varepsilon = 0.02$). (d) The CPO intensity is quantified as the M-index from the ODF (misorientation index, Skemer et al., 2005) and its evolution with cumulative strain so shown. The evolution of the CPO in simple shear only, starting with a random fabric (single-regime deformation), is shown in green in both (c) and (d).

At the beginning of the second regime the CPO slightly rotates antithetically (*i.e.*, opposite to the imposed shear sense) (see $\varepsilon_2=1$ in Fig. 3b), while the CPO intensity (Fig. 3d) and girdle component decrease (Fig. 3c). After that the CPO intensity (point maximum) gradually intensifies again and the *c*-axes align at about 10° to the normal to the shear plane (see $\varepsilon_2=3$ in Fig. 3b and Fig. 3c). After the two deformation events the resulting CPO resembles the one resulting from simple shear deformation only, in both CPO intensity and symmetry (green and blue markers in Fig. 3d and Fig. 3c). This implies that the previous uniaxial flattening regime is not recognizable after only a moderate amount ($\varepsilon_1>1$) of subsequent simple shear, as may be expected as the change in CPO only involves a small rotation and intensification of the *c*-axes point maximum.

### 4.2. Series B: Ice flowing from the centre of the ridge to deep lateral zones

On ridges away from ice domes, ice flows away and stretches in the direction perpendicular

to the ridge (Wang et al., 2002; Faria et al., 2014) (zone I in Fig. 1). Here we consider the end-member case that this leads to uniaxial extension in the flow direction. Similar to Series A, the ice is then assumed to be buried and to enter zone II (Fig. 1) where flow is dominated by bedrock-parallel simple shear, again in the direction perpendicular to the ridge (zone II in Fig. 1). For this case, the initial microstructure is first deformed by **V₃**, horizontal uniaxial extension parallel to *x*, followed by **V₂**, dextral simple shear in the vertical plane (*xz*) (Tables 1 and 2). A sensitivity test of series B, where the first deformation regime applied is pure shear (**V'₃** in table 1) is shown in supplementary figure 1.

In series B, the initially random distributed *c*-axes develop a CPO characterised by a strong vertical girdle, in coherence with the horizontal extension applied during the first deformation regime (see $\varepsilon_1$=0.92 in Fig. 4a). From the final step onwards, the aggregate is deformed under simple shear in the vertical plane (*xz*) up to a natural strain of $\varepsilon_2$=4 (Fig. 4b). *C*-axes reorient following the new flow, destroying the vertical girdle fabric (Fig. 4c), and forming a broad single point maximum almost normal to the shear plane (see $\varepsilon_2$=4 in Fig. 4b). Although the final CPO symmetry is coherent with simple shear deformation, its shape after a strain of $\varepsilon_2$=4 still differs from that of the previous case (series A) or that of simple shear only (see the last step for the second regime in Figs. 3a and 4a). At the highest modelled strain ($\varepsilon_2$=4) the CPO intensity is clearly lower than that developed under solely simple shear deformation (Fig. 4d). Changing a girdle to point maximum requires a significantly higher strain than the formation of a point maximum only (green versus blue graph in Fig.4d).

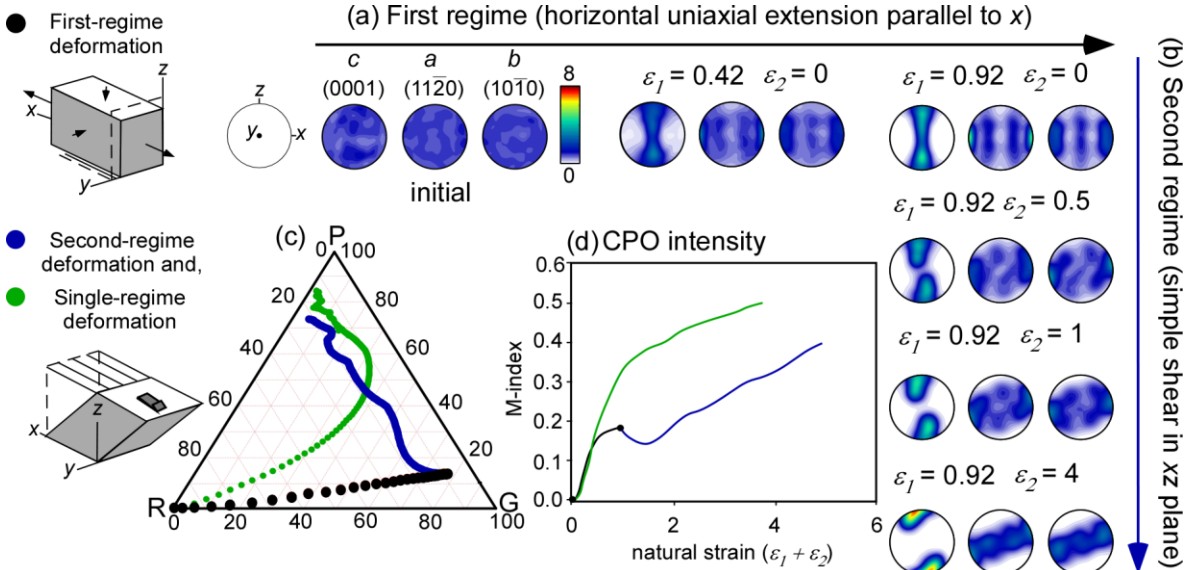

**Figure 4.** CPO evolution during the two consecutive deformation regimes of series B. The final texture at the end of the first flow regime (a), is used as initial texture for the second regime (b). (c) Ternary diagram that

shows the CPO symmetry expressed as the proportion of point (P), girdle (G) and random (R) components of $c$-axes (0001). Points plotted at the PGR graph represent every step of deformation ($\varepsilon = 0.02$). (d) The CPO intensity is quantified as the M-index from the ODF. The evolution of the CPO in simple shear only, starting with a random fabric (single-regime deformation), is shown in green in both (c) and (d).

**4.3. Series C: Ice flowing from an ice dome to an ice flank or stream**

In this configuration we assume that ice is first gradually flattened by vertical uniaxial compression parallel to $z$ in an ice dome ($\mathbf{V_1}$; zone I in Fig. 1) and subsequently enters a zone
in which the flow accelerates in, for example, an ice stream, leading to uniaxial extension in the flow direction ($\mathbf{V_3}$; zone III in Fig. 1). Series C includes a set of simulations where $\mathbf{V_3}$ deformation starts at different strains in the first stage with $\mathbf{V_1}$. A sensitivity test of series C, where the second deformation regime applied is pure shear ($\mathbf{V'_3}$ in table 1) is shown in supplementary figure 2.
The microstructure in series C first develops a strong CPO (Fig. 5b) with a point maximum in the $z$-direction of the axial compression applied during the first deformation regime (see first row in Fig. 5a). In this series the second deformation regime was applied at different steps of natural strain of the first regime (at $\varepsilon_1$=0.2, 0.4, 0.8 and 1.2, respectively). The intensity of the initial deformation regime has a notable influence on the CPO developed during the second
regime. In the absence of first-regime deformation ($\varepsilon_1$=0.0), where the microstructure only experiences uniaxial extension in the $x$-direction, a vertical girdle fabric develops (see first column in Fig. 5a and green marker Fig. 5b). Regardless of the step at which the second regime is applied, the CPO tends to evolve towards a girdle fabric (see evolution of all series in Fig. 5a-b), which slightly reduces the CPO intensity of the start of the second regime (Fig.
5c). If the second deformation regime is applied when the microstructure has only been slightly affected by the first regime ($\varepsilon_1 < 0.4$), the $c$-axes tend to rotate forming a final girdle fabric but with a recognizable point maximum component, resulting in an "hourglass shape" (see second and third columns in Fig. 5a). However, if a strong point maximum CPO developed during the first deformation regime ($\varepsilon_1 > 0.8$), the second regime cannot modify
the inherited $c$-axis orientation at a natural strain of $\varepsilon_2 = 1.2$ (see fifth column in Fig. 5a), and the final CPO continues to be dominated by a strong point maximum (Fig. 5b). The effect of the second regime is observable in the $a$-axis distribution, because regardless of the intensity of the first regime, the extension along the x direction produces three $a$-axis maxima (see last row in Fig. 5a). The result suggests that converting a point maximum into a girdle takes even

more strain than the opposite (Series B).

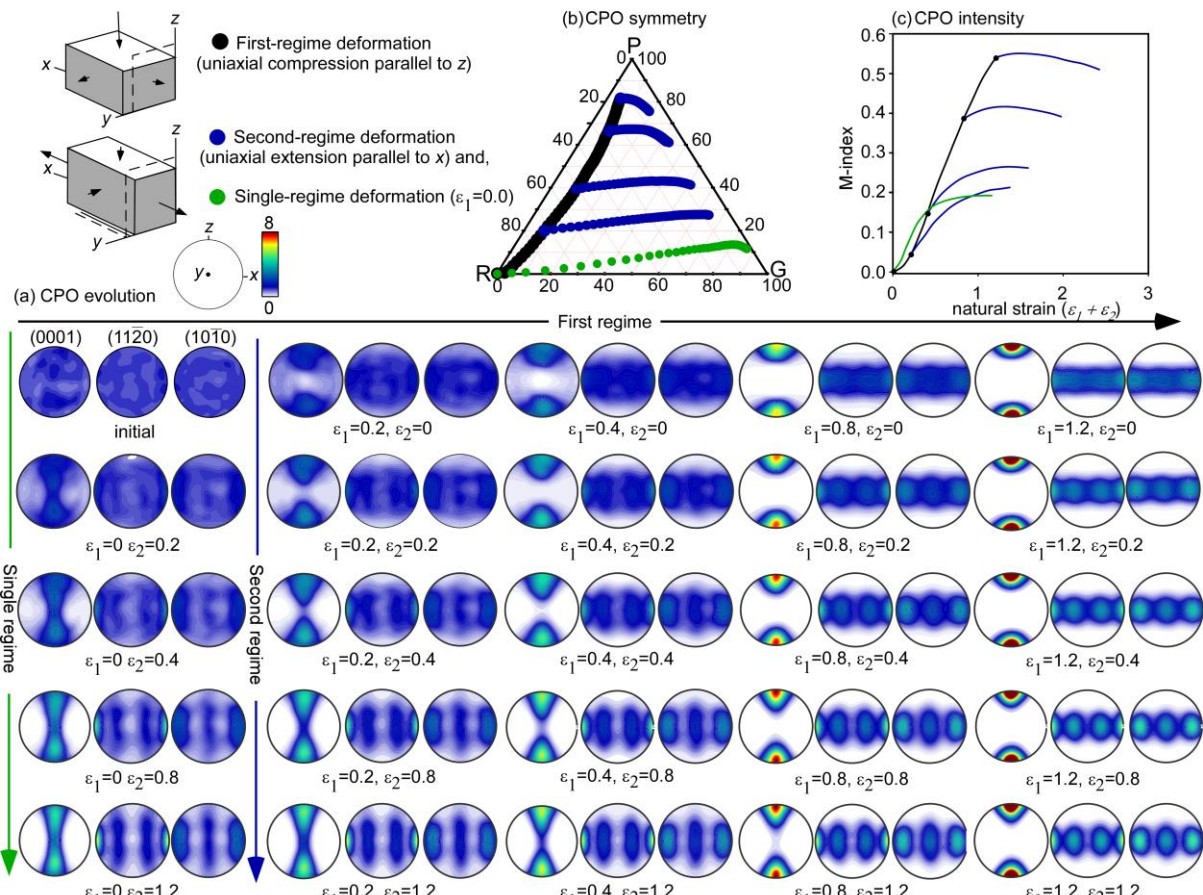

**Figure 5.** (a) CPO evolution during two consecutive deformation regimes of series C. Textures developed at
different steps of natural strain in the first regime ($\varepsilon_1$) are used as initial textures for the second regime ($\varepsilon_2$). (b)
Ternary diagram that shows the CPO symmetry expressed as the proportion of point (P), girdle (G) and random
(R) components of $c$-axes (0001). (c) The CPO intensity is quantified as the M-index from the ODF. The
evolution of the CPO in uniaxial extension parallel to $x$ only, starting with a random fabric ($\varepsilon_1$=0: single-regime
deformation), is shown in green in both (b) and (c).

## 4.4. Series D: Ice flowing from an ice-stream or glacier to an ice shelf or shear margin

Dominantly simple shear deformation in the horizontal plane takes place in the margins of ice
streams (Hudleston, 2015) and can be found in some ice shelves (Young et al., 2002; LeDoux
et al., 2017; Lutz et al., 2020) (zone IV in Fig.1). Ice affected by this shearing may have
experienced different deformation types before. Here we consider the case of uniaxial
extension in the flow direction (zone III in Fig.1), which would represent the ice above the
deepest layers that thus did not experience bedrock-parallel simple shear. This series include

a first deformation regime of uniaxial extension in the *x* direction (**V₃**), and subsequently

dextral simple shear in the horizontal plane (*xy*) (**V₄**). A sensitivity test of series D, where the

first deformation regime applied is pure shear (**V'₃** in table 1) is shown in supplementary

figure 3.

In series D a vertical girdle fabric develops during the first deformation regime ($\varepsilon_1 = 0.92$ in

Fig. 6a), which then rapidly evolves towards a point maximum due to simple shear in the

horizontal plane (*xy*) imposed during the second flow regime (Fig. 6b). However, after

significant strain there is still a girdle component.

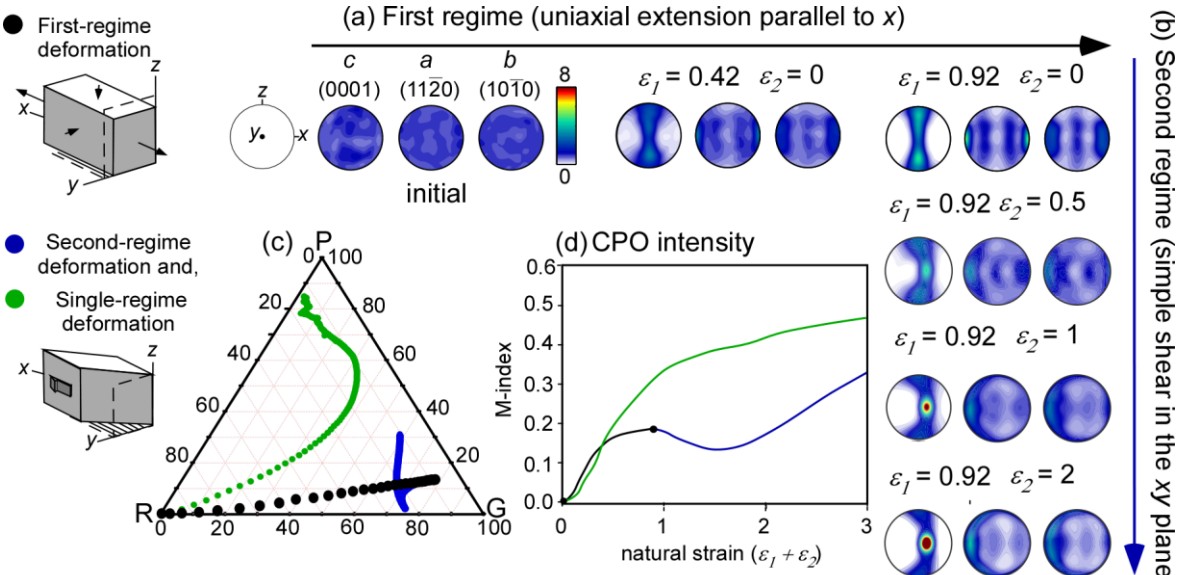

**Figure 6.** CPO evolution during the two consecutive deformation regimes of series D. The final texture at the

end of the first regime (a), is used as initial texture for the second regime (b). (c) Ternary diagram that shows the

CPO symmetry expressed as the proportion of point (P), girdle (G) and random (R) components of *c*-axes

(0001). Points plotted at the PGR graph represent every step of deformation ($\varepsilon$=0.02). (d) The CPO intensity is

quantified as the M-index from the ODF. The evolution of the CPO in simple shear only (single-regime

deformation), starting with a random fabric, is shown in green in both (c) and (d).

## 5. Discussion

Experiments and simulations with an initial random distribution of crystallographic

orientations predict a quick alignment of the crystallographic axes according to the imposed

deformation conditions (Azuma and Higashi, 1984; Qi et al., 2019, Llorens et al., 2016a;

Llorens et al., 2016b; Peternell and Wilson, 2016; Wilson et al., 2020). The types and

orientations of CPOs are best described in terms of the principal axes of the finite-strain

ellipsoid (FSE), also known as finite stretching axes (FSA's) that are typically labelled *X, Y*

and *Z* from longest to shortest (Fig.7) (Passchier, 1990). These axes are parallel to the instantaneous stretching axes (ISA's) in the case of coaxial deformation, but not in the case of simple shear. In simulations, *c*-axes tend to align themselves parallel to the shortening FSA's. In the case of uniaxial compression ($V_1$ in Table 1*; X=Y>Z*) this leads to the development of a strong point maximum. The development of a vertical single maximum observed in vertical uniaxial compression simulations performed (first-regime deformation in series A and C) match observations from Antarctic natural ice core samples, as the EPICA Dome C (Durand et al., 2007), Dome F (Azuma et al., 1999), Talos Dome (Montagnat et al., 2012), and Greenland ice cores GISP2 (Gow et al., 1997) and GRIP (Thorsteinsson et al., 1997). They also match laboratory experiments carried out at low temperatures and high strain rates (Craw et al., 2018; Fan et al., 2020). A girdle forms in uniaxial extension ($V_3$ in Table 1; *X>Y=Z*). The vertical girdle fabric predicted in our simulations under uniaxial extension (first-regime deformation in series B and D) is also observed in Antarctic and Greenland ice cores, such as the Vostok ice core (Lipenkov et al., 1998; Voigt, 2017), EPICA DML (Weikusat et al., 2017), NEEM (Montagnat et al., 2014a) or in the Styx Glacier in Northern Victoria Land (Kim et al., 2020). As described in the NorthGRIP Greenland ice core, the vertical girdle is interpreted as evidence for extension transverse to the ridge, where the vertical girdle plane is oriented perpendicular to the axis of horizontal extension (Wang et al., 2002; Faria et al., 2014; Weikusat et al., 2017).

In simple shear ($V_2$ and $V_4$) the orientation of the *Z*-axis rotates from initially 45° to 0° to the shear plane at infinite strain, while the *X*-axis remains in the shear plane. A point-maximum CPO is expected and observed parallel to *Z* in the plane between the instantaneous shortening axis and the normal to the shear plane. *a*-axes (11-20) align themselves perpendicular to the *c*-axes in the plane of the two least compressive FSA's (*i.e.* the *X* and *Y*-axes of the finite-strain ellipsoid) (see first column in Fig. 7).

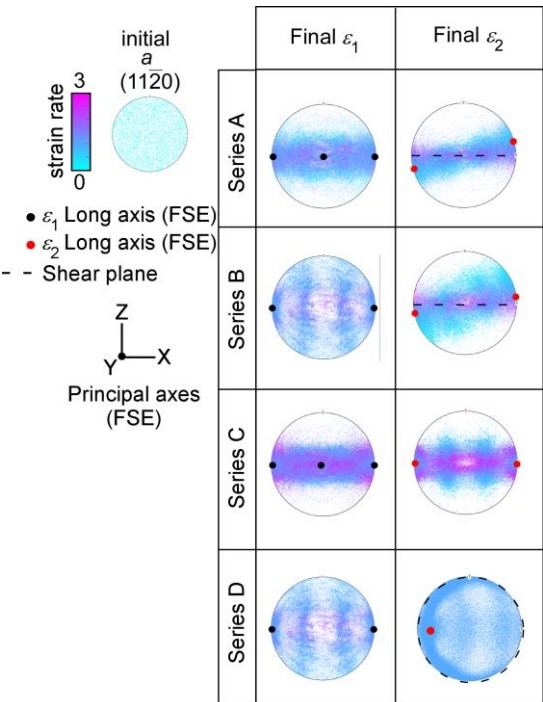

**Figure 7.** Pole figures of *a*-axis (11-20) orientation at the end of the first stage of deformation ($\varepsilon_1$) and the beginning and the end of the second stage of deformation ($\varepsilon_2$) for all series presented. The colour code in pole figures indicates the Von Mises strain rate normalised with respect to the bulk value per point (*unode*). The orientation of the elongation axis of the finite-strain ellipsoid (FSE) is indicated in solid black or red dots for the first and second stage of deformation, respectively. The shear plane in simple shear simulations is plotted as a black dashed line. When the maximum elongation axis lies within the *a*-axis girdle (series C and D), the inherited CPO is not completely overprinted at the end of the simulation.

The results from plausible scenarios studied here reveal that when a second deformation regime affects the microstructure the inherited CPO gets overprinted, and the CPO evolves according to the new kinematics of deformation. Regardless of the inherited CPO, our simulations predict a strong *c*-axis maximum almost perpendicular to the shear plane under simple shear boundary conditions (series A and B). In both series, the *a*-axes reorient following the elongation axis of the new regime (*i.e.*, long axis of the finite-strain ellipsoid, FSE) with high-strain rate domains oriented parallel to the shear plane (see high-strain orientations in series A and B in Fig. 7). Following these results, the CPO developed in ice accumulated in a dome or a ridge would be overprinted at depth by the dominance of shearing parallel to the bedrock. Considering strain rates as at modelled the base of the EDML ($\sim 9 \times 10^{-11} \mathrm{s}^{-1}$ ; Weikusat et al., 2017) the CPO would be preserved for at most $\sim 2.8$ kyr. These predicted CPOs are in accordance with the observations of CPO analyses from most deep ice drill cores located on sites with the ice frozen to the bed in Antarctica and

Greenland, where an approximately vertical single maximum CPO is found (*e.g.*, Faria et al., 2014; Montagnat et al., 2014a; Weikusat et al., 2017).

Simulations predict that the land-based ice CPO is destroyed at the ice shelf, assuming the example of shearing in the horizontal plane (*xy*) as the dominant deformation regime, as observed in the Antarctic Amery (Young et al., 2002) or Western Ross ice shelfs (LeDoux et al., 2017; Lutz et al., 2020). In this case, although the inherited vertical girdle CPO is not completely destroyed at the end of the simulation, it is progressively overprinted by the shearing in the horizontal plane, (series D in Fig. 7). Assuming strain rates calculated in ice shelves ($\sim 5\text{x}10^{-11}\text{s}^{-1}$ ; Millstein et al., 2021) the inherited fabric would be preserved for at most $\sim 2.5$ kyr.

When an inherited single vertical maximum is affected by extension along the flow, similar to ice flowing in an ice stream that previously accumulated in an ice dome, the CPO is also progressively overprinted, but the required strain is clearly much higher than that for the other cases ($\varepsilon > 1.2$) (series C Fig. 7). Simulations predict that ice moving into a shear margin or potentially developing a shear margin during the initiation of an ice stream, will basically lose its original CPO as shearing in the horizontal plane (*xy*) overprints the inherited fabric. As observed in the simulations, the effectiveness of the second flow regime on the reorientation of the CPO depends not only on the strength of the inherited CPO, but also on the relationship between the crystal orientation and the finite-strain ellipsoid of the new imposed regime. When the maximum elongation axis of the new finite-strain ellipsoid lies within the inherited *a*-axis girdle (see series C and D in Fig. 7) *c*-axis orientations are more slowly modified. This is in accordance with our understanding of dislocation motion, because the most efficient dislocations in ice (basal dislocations) do have a Burgers vector component (glide direction) along the *a*-axes.

The flow behaviour from the pole figure observed in Series C at $\varepsilon_1 = 1.2$ and $\varepsilon_2 = 1.2$ (see lower right corner in Fig. 5a) could easily be misinterpreted as axial compression in the *z*-direction (Fig. 3b), while the true current flow is axial extension in the *x*-direction.

The influence of the second flow regime on the reorientation of the inherited CPO is also observable in the relative activity of slip systems (Fig.8). In all cases, the new imposed regime produces a remarkable increment of basal slip, while the pyramidal slip system activity is reduced (see series A, B and D in Fig.8). A different effect is found when an inherited vertical single maximum is affected by extension along the flow direction, where the basal activity remains constant, and the prismatic slip is increased (see series C in Fig.8).

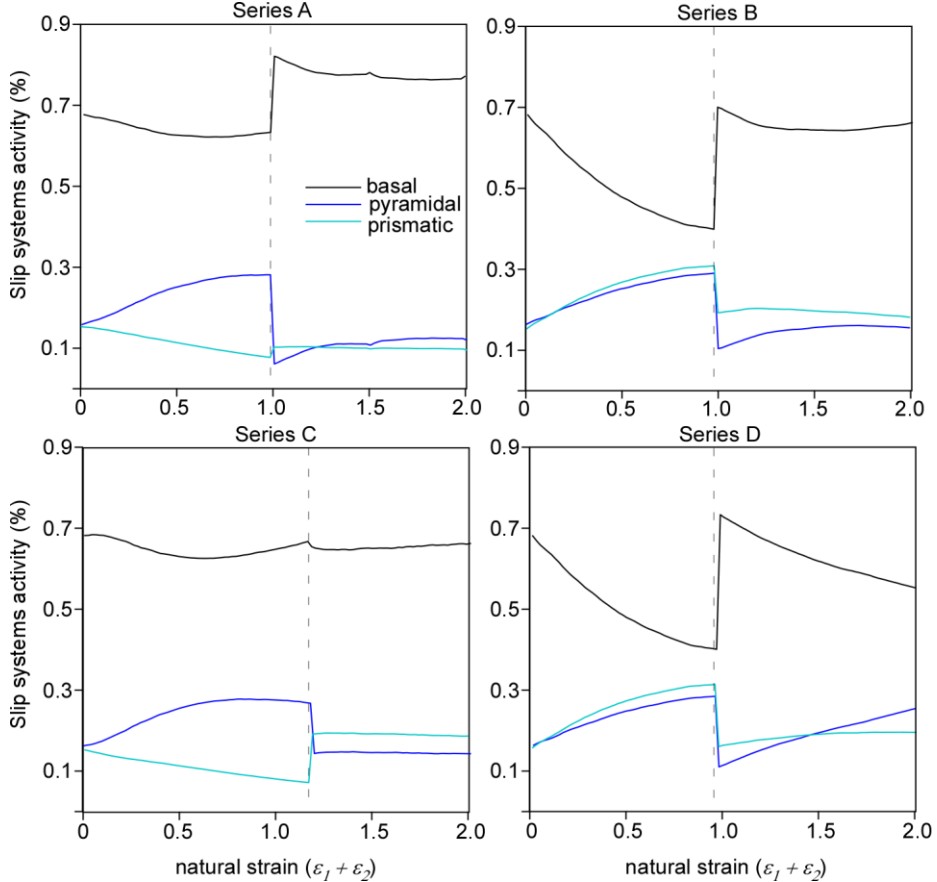

**Figure 8.** Evolution of the relative activities of basal, pyramidal and prismatic slip systems during deformation for all series presented, calculated from Equation (1). Transition of deformation regimes are marked with a vertical dashed line. In A, B and D series, the second flow regime produces a remarkable increment of basal slip, while the pyramidal slip system activity is reduced. However, in series C the basal activity remains

constant, and the prismatic slip is increased.

All these results demonstrate that an inherited CPO can change entirely within strains of $\varepsilon$ <1.2 with a range of transition fabrics, with the exception of series C, when a strong point maximum CPO developed during the first deformation regime (Fig.9). Our results suggest

that, under natural conditions, as for example those at the onset of the NEGIS, where the velocity increases by 40 m/yr over a distance of 120 km (i.e., strain rate of $\sim 1 \times 10^{-11}$ s$^{-1}$) (Joughin et al., 2018) an inherited fabric would be preserved for at most $\sim$ 7 kyr. However, in the NEGIS margins, where strain is considerably higher (i.e., strain rate of $\sim 4 \times 10^{-10}$ s$^{-1}$) (Joughin et al., 2018), an inherited fabric would be destroyed in a mere $\sim$ 200 years. This

process depends on both the intensity of the inherited CPO, and its orientation with respect to the new flow regime. These results are in agreement with ice experiments of natural samples with a pre-existing CPO, where the application of a stress field in a non-favourable orientation with respect to the inherited CPO destroys it (Jun and Jacka, 1998, Craw et al.,

2018).

One particular case is an inherited strong CPO, where *a*-axes are optimally oriented with respect to the second flow regime, because the finite-strain ellipsoid elongation axis from the first and second regime are in the same plane (see FSA in series C and D in Fig.8). In this situation the second regime does not produce a rapid re-orientation of the *a*-axes with respect to the elongation axis, the basal activity is not enhanced and therefore the CPO is slowly modified (series C and D in Fig.9). In this case the required finite strain for overprinting the initial CPO is clearly much higher than that for the other cases.

The entire change of a previous CPO also takes place in other rocks, such as olivine-rich rocks in the upper mantle, where a new CPO quickly develops according to the new imposed boundary conditions. The observed CPO will thus not record the full history of changes in the kinematics of deformation (Kaminski, 2004). However, as our results reveal, the re-orientation of an inherited CPO depends both on its intensity and on the orientation with respect to the new stress field. These results are in agreement with observations from olivine experiments, where the pre-existing texture orientation determines the way the texture evolves (Boneh and Skemer, 2014). Accordingly, the deformation history could have an impact on the CPO in areas with complex flow, as in subduction zones (Di Leo et al., 2014; Li et al., 2014).

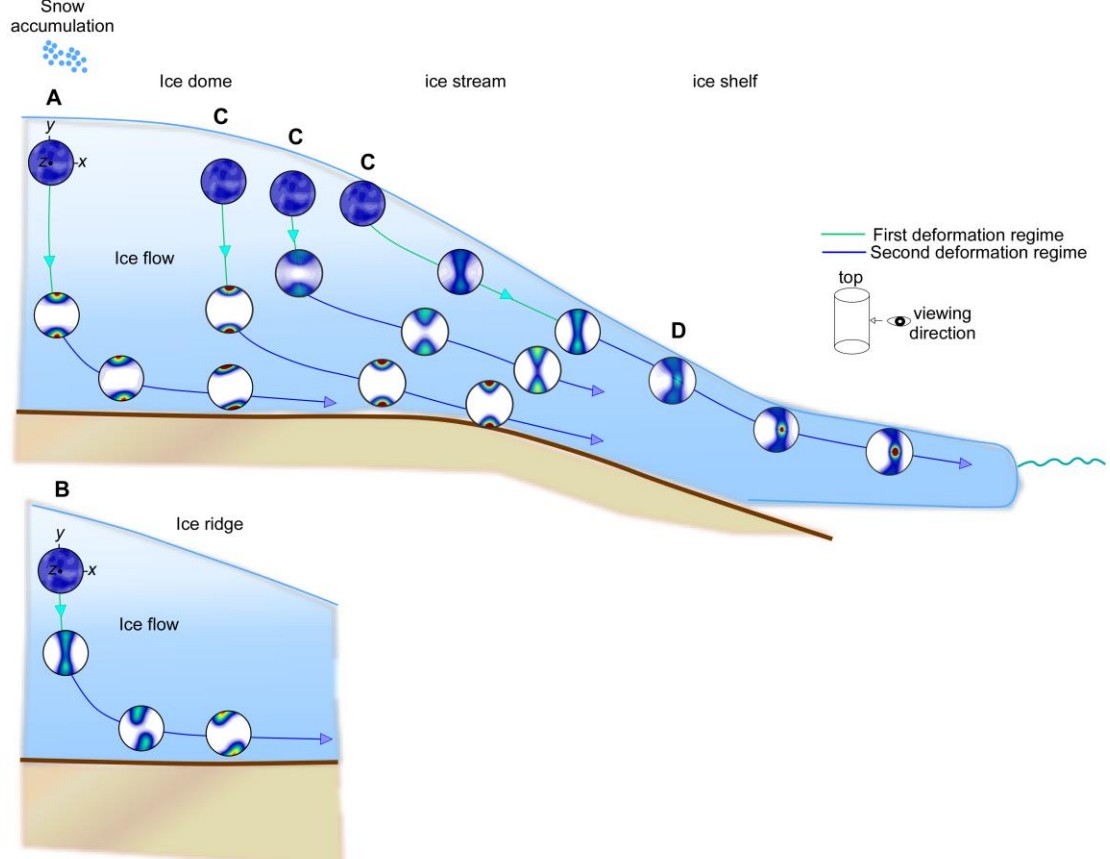

**Figure 9.** Prediction of evolution of *c*-axis (0001) orientation in all series presented (from A to D), according to the deformation conditions assumed in figure 1. A flow change produces an overprint of the previous CPO, with a range of transition fabrics.

## 6. Model limitations and further processes

The models presented consider only deformation and exclude recrystallisation processes. This needs some discussion as recrystallisation is inferred to be an important process in both laboratory experiments (Fan et al., 2020; Journaux et al., 2019; Kamb, 1972; Montagnat et al., 2015; Qi et al., 2017; Qi et al., 2019) and in nature (Duval and Castelnau, 1995; Gerbi et al., 2021; Jackson and Kamb, 1997; Monz et al., 2021; Thomas et al., 2021). There are three aspects to assess when considering the limitations of excluding recrystallisation processes from model outcomes:

1. Do dynamic recrystallisation processes, that we can model, significantly change the modelled CPO patterns (symmetry, orientation, strength)?

2. Are there key processes that may affect CPOs and that cannot yet be modelled?

3. Are model representations of recrystallisation kinetics robust:? do recrystallisation processes change the rate (as a function of strain or time) at which CPOs develop?

A number of studies that use the VPFFT-ELLE modelling approach (e.g., in Llorens et al., 2016a, 2016b and 2017; Steinbach et al., 2017 and Gomez-Rivas et al., 2017) indicate that model representations of dynamic recrystallisation processes, including grain boundary migration, subgrain rotation, intracrystalline recovery and polygonization, have a minor effect on the CPO pattern development (symmetry, orientation, strength). For the current series of simulations, we tested this (suppl. Fig. 4) and again found this to be the case. Including recrystallisation processes that we are able to model will make little difference to model outcomes while making the simulations more complex.

Some elements of the effect of recrystallisation on CPOs are not well captured in detail when these processes are modelled. These include small circle girdles in compression and double maxima in shear. Experiments conducted at conditions where dynamic recrystallisation dominates (relatively high temperatures and low strain rates) give small circle girdle CPOs under uniaxial compression (Jacka and Maccagnan, 1984), with the small circle closing and becoming a weak maximum parallel to compression at conditions where dynamic recrystallisation is reduced relative to deformation (Fan et al., 2020; Qi et al., 2017). Double maxima tend to be developed in shear where dynamic recrystallisation dominates (Qi et al., 2019). Although developing ways that better capture these recrystallisation effects is important (e.g. Richards et al., 2021), the effect on CPO patterns are details that are unlikely to significantly affect the analysis carried out in the present study.

Grain boundary sliding (GBS) is a process that is not included in our simulations, although some studies suggest it may play a role in natural ice flow (Fan et al., 2020; Behn et al., 2021). Its effect on microstructure and CPO is, however, not well established. It is suggested that it would reduce the strength of a CPO (Richards et al., 2021) and experiments tentatively support this (Craw et al., 2021; Fan et al., 2020), but we are not aware of any study showing that the CPO would be significantly altered otherwise. From this we deduce that, if GBS would operate, our results would probably still largely apply, except for the strength of the CPOs that were modelled without GBS.

The effect that recrystallisation has on rates of CPO change is largely unconstrained. Duval and Castelnau (1995), estimate that microstructures of polar ice can be entirely recycled with little strain (by 1% strain at -10ºC and natural strain rates). It is not clear how the microstructural re-organisation corresponds to CPO modification. There are very few attempts in experiments to change one CPO to another (Craw et al., 2019 is the only one we know of). And there are no field studies where ice is collected on transects where the deformation kinematics change along the transport pathway so that CPO change rates can be

documented. Thomas et al (2021) show CPOs that relate kinematically to a marginal shear zone, in ice where they infer that the grain microstructure (size and shape) has been overprinted by the effects of tidal flexure deformation. If this is right, this is an example of recrystallisation changing grains but not CPOs. Understanding the kinetic effects of recrystallisation is an important area of research for the future.

**7. Conclusions**

This study presents a series of full-field numerical simulations of non-linear viscous polycrystalline ice deformed under two consecutive flow regimes. The analysis of the
590 resulting crystallographic preferred orientation (CPO) leads to the following main conclusions:

1. Simulations with an initial random distribution of crystallographic orientations predict a quick alignment ($\varepsilon$< 0.4) of the crystallographic axes according to the imposed boundary conditions. In all cases, ice polycrystals develop a CPO with $c$-axes mostly oriented parallel
to the compression axis, and $a$-axes oriented parallel to the elongation axis of the finite strain ellipsoid.

2. Depending on the sequence of deformation regimes, an inherited CPO can be completely overprinted by the later deformation event. An inherited CPO can change entirely within a natural strain lower than 1.2, with a range of transitional fabrics. This process depends both
on the intensity of the inherited CPO and its orientation with respect to the new stress field.

3. More specifically, when the inherited $a$-axis preferred orientation and the elongation axis of the finite strain ellipse of the flow are parallel, lattice orientation only needs to be reoriented slightly. The required finite strain for overprinting the initial CPO in this configuration is much higher ($\varepsilon$ >1.2) than that for the other cases. This is the case of ice
flowing from an ice dome to an ice stream. This situation could lead to a misinterpretation of the second flow regime from the observed $c$-axis preferred orientation.

4. According to our results, CPOs are reliable indicators of the current flow conditions, as they usually adapt to them in a relatively short time. However, caution is warranted when a volume of ice may have experienced consecutive flows  with the extension direction in the
610 same direction

**Acknowledgements**

MGL acknowledges a Juan de la Cierva-Incorporación fellowship (IJC2018-036826-I), funded by the MCIN/AEI/10.13039/501100011033. EGR acknowledges the support of the Ramón y Cajal fellowship (RYC2018-026335-I), funded by the MCIN/AEI/10.13039/501100011033 and the FSE. IW acknowledges the HFG grant no.VH-NG-802. This work is part of the CSIC-PTI POLARCSIC activities, and it has been developed using the facilities of the Laboratory of Geodynamic Modelling of GEO3BCN and a computer cluster of the University of Tübingen. Funding was provided partly by GeoCAM (PGC2018-095154-B-100), Spanish Government.

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
