# Peer review of "Can changes in deformation regimes be inferred from crystallographic preferred orientations in polar ice?"

_The Cryosphere, 2021_

## Referee Comment (RC1)

**Review of "Can changes in ice-sheet flow be inferred from crystallographic preferred orientations?" by Llorens et al.**

This study uses a full-field model to examine the overprinting of ice-crystal fabrics when two strain regimes are experienced. The aim is to understand how/whether the fabric records the ice-flow history or if it is rather representative only of the in-situ deformation regime. Contrary to the title, I do not think it really evaluates "changes in ice-sheet flow," since simulations intentionally mimic idealized steady-state conditions. Nonetheless, this is an interesting topic that has received other attention in the last year, but this study uses a different approach that likely represents the fabric evolution more accurately. The paper is well organized and the writing is clear, with the exception of some unusual notation that created unnecessary confusion for me. The figures are exceptionally well done (Figure 9 in particular could be used to teach good science communication). The manuscript is relevant for *The Cryosphere* and it could be a valuable contribution once some important issues are addressed.

**Specific comments:**

1. I think a bit of consideration is needed surrounding what it means to "infer changes in ice-sheet flow," since the present manuscript does not actually address that question. Fundamentally, all simulations here are steady state; they follow idealized particle paths within a steady-state ice sheet, though of course those particle paths transit multiple strain regimes. Since we may be curious whether the fabric matches the in-situ conditions, the paper could simply be retitled to something like "Do crystallographic preferred orientations always represent in-situ conditions?" with the body essentially left as-is.

   If the authors are instead set on addressing a question about flow changes (which is probably more interesting and relevant for a broader audience), then I think more simulations, as well as some explanation of what that would entail, is needed. I would think some kind of change to flow is needed in the simulations to infer a flow change (i.e. not a change along a particle path, but a change to the large scale flow through which the simulated parcel transits). For example, what does the model say about a transition from a dome to a ridge? How long/over what strain would such a change be evident in the CPO? How about formation of an ice stream? Along with those simulations, extensive evaluation like in line 440 would be warranted (i.e. do those changes manifest unambiguously in the CPO? what could we see in the CPO that allows inference of a flow change?).

2. The results are hard to believe until dynamic recrystallization (DRX) is given more consideration. The two citations used to justify its exclusion are both modeling studies that in my view are outliers compared to the conventional wisdom on the effect of migration recrystallization on crystal fabric from ice cores (Faria et al., 2014a), experiments (Fan et al., 2021; Qi et al., 2019; Journaux et al., 2019), and modeling (Richards et al., 2021; Faria et al., 2014b). Migration recrystallization is often described as depending on the stress rather than the strain (Duval and Castelnau, 1995), and so may be particularly relevant for $V_2$ and $V_4$ (near the bottom of the ice sheet or in shear margins) where stresses are presumably high. Moreover, even if we were to assume that the effect of recrystallization were relatively small, why does excluding it better represent how CPO responds to a flow change (as implied by the current version of the manuscript)? This concern is intensified because this study shows that, under lattice rotation, development of the new fabric is strongly dependent on the previous fabric, so might a similar sensitivity apply to DRX? This issue is critical; if recrystallization changes the timescale/strain scale over which fabric persists, then a model of lattice rotation alone cannot accurately capture whether flow history can be inferred (or even whether the fabric matches in the in-situ stress and strain). I think this issue is sufficiently important that consideration of

different mechanisms of DRX is needed (i.e. rotation and migration recrystallization). The large strains needed to overprint fabric seem to depend on the precise misorientation of the crystallographic axes relative to the new strain, and it seems plausible that even minor effects of rotation recrystallization could alter this misorientation and thus change the results, even if migration recrystallization does not lead to strong CPOs.

3. I do not think that $V_3$ is an accurate representation of a ridge. Almost by definition, a ridge experiences confined compression/extension rather than pure extension, so the deformation gradient at the ridge itself is

$$\nabla u = \begin{pmatrix} a & 0 & 0 \\ 0 & -a & 0 \\ 0 & 0 & 0 \end{pmatrix}$$

for some $a$. Of course, some areas can have flow convergence as ice leaves the ridge, in which case we have something like

$$\nabla u = \begin{pmatrix} a+b & 0 & 0 \\ 0 & -a & 0 \\ 0 & 0 & -b \end{pmatrix}$$

but to my knowledge $b < a/2$ in such areas; the same would be true for ice streams. The

$$\nabla u = \begin{pmatrix} a & 0 & 0 \\ 0 & -a/2 & 0 \\ 0 & 0 & -a/2 \end{pmatrix}$$

used in the manuscript will have a greater tendency to form a girdle since the extensional stress is equal in all directions in a vertical plane. Because this may affect the results, I would like to see series B, C, and D redone with more realistic conditions, or at least a sensitivity test with

$$\nabla u = \begin{pmatrix} a & 0 & 0 \\ 0 & -a & 0 \\ 0 & 0 & 0 \end{pmatrix}$$

Along these lines, I am a bit skeptical of the total strains experienced with $V_4$ as the second condition. Is there anywhere where ice spends long enough in a shear margin to reach these total strains? On a particle path, I would expect the particle to enter the ice stream or shelf before such high strains are reached. I do not see this as key to the results overall, so just a sentence mentioning whether it is realistic may be sufficient.

4. Although strain is the classical scale for fabric development, most glaciologists do not think in terms of total strain when working on problems other than fabric. To make the work more accessible, it would therefore be nice to give numbers as approximate timescales as well (I assume this is easy since the strain rates are known). It would also be nice to say something along the lines of "under realistic conditions, CPO can be preserved for XX years, and a flow change YY years ago could be detected."

**Technical corrections:**
L62: I do not think this is the intended Alley paper—perhaps (Alley, 1988)?
L95: There are two other studies that model fabric changing in new deformation regimes.
(Thorsteinsson et al., 2003) looked at some examples with overprinting. This exact question is

addressed by (Lilien et al., 2021). These studies do not negate the novelty of the present work, since they both used a different type of model, but this is not the first study to consider such a question.

L116: This quite circuitous—we have direct measurements of velocity that show extensional flow at ridges (or, really, we only call them ridges because flow is extensional), so there is no need to use CPO to conclude it.

L137: I strongly suggest altering the notation. $z$ as the vertical coordinate in 3D is such a widespread convention that using $y$ vertical leads to unnecessary confusion, and I see no benefit. This is compounded by the terminology for shear; the authors refer to the shear plane rather than the plane in which the shear happens (e.g. "horizontal simple shear" for shear in their xy), which I have heard called "vertical simple shear" since motion differs vertically. I suggest calling it "simple shear in the vertical plane" to avoid all ambiguity. I am particularly confused by things like line 225, where the authors call $V_2$ shear "on" the horizontal plane (I think this is a typo, but the terminology leaves me unsure).

L173: Perhaps I misunderstand how the full-field model works, but why is the bulk exponent discussed here? The model captures individual grains, so should we not care about the grain exponent, which need not be the same as the bulk exponent (e.g., Rathmann et al., 2021)? Experimental evidence for slip on individual monocrystal slip systems indicates that the exponent is in fact closer to 2 for basal glide (Duval et al., 1983 Figure 2), and I do not think this is evaluated in Bons et al. To be clear, I do not think that anything in the simulations needs to change, but it would be good if this discussion clarified the grain/bulk distinction and did not introduce the bulk n=4 discussion unless needed.

Eq 1: This equation should be re-written to conform with standard typesetting conventions, and the explanation should be expanded. What is the summation variable $S$ (I assume it is the slip system)? I am guessing that $sgn$ is the sign, but by convention (and ISO standard) that should be non-italic (indeed, I spent a while wondering why gravity, n and s were being multiplied). I also suggest dropping the "x" for multiplication, as with tensorial quantities it is often confused with the cross product; ISO standards allow skipping the symbol entirely.

~L205: There is no mention of the single-regime simulations that I can see—it would be good to mention these in the first paragraph.

L455: This does not seem like a fair characterization of Smith et al., 2017; I do not think they claimed anything that contradicts the results here. They note that large flow changes have occurred recently in ice streams and argue that this may be evident in the fabric. As pointed out in the specific comments, this it would be useful to put a timescale on the results here as well as a natural strain so that results can be compared to other studies. My sense is that there is no conflict, but regardless to dismiss their consideration of the possibility by calling it an "assumption" is inaccurate.

L464: This makes it sound like the CPO does not change, when I think the point is intended to be that it changes much more slowly.

Figure 8: "with vertical dashed line."

L532: FSE was previously defined

L478: This seems to be a conclusion of previous studies rather than the present one

**References:**

Alley, R. B.: Fabrics in polar ice sheets: development and prediction., Science, 240, 493–5, https://doi.org/10.1126/science.240.4851.493, 1988.

Duval, P. and Castelnau, O.: Dynamic Recrystallization of Ice in Polar Ice Sheets, J. Phys. IV, 05, C3-C3-205, https://doi.org/10.1051/jp4:1995317, 1995.

Fan, S., Prior, D. J., Cross, A. J., Goldsby, D. L., Hager, T. F., Negrini, M., and Qi, C.: Using grain boundary irregularity to quantify dynamic recrystallization in ice, Acta Mater., 209, 116810, https://doi.org/10.1016/j.actamat.2021.116810, 2021.

Faria, S. H., Weikusat, I., and Azuma, N.: The microstructure of polar ice. Part I: Highlights from ice core research, J. Struct. Geol., 61, 2–20, https://doi.org/10.1016/j.jsg.2013.09.010, 2014a.

Faria, S. H., Weikusat, I., and Azuma, N.: The microstructure of polar ice. Part II: State of the art, J. Struct. Geol., 61, 21–49, https://doi.org/10.1016/j.jsg.2013.11.003, 2014b.

Journaux, B., Chauve, T., Montagnat, M., Tommasi, A., Barou, F., Mainprice, D., and Gest, L.: Recrystallization processes, microstructure and crystallographic preferred orientation evolution in polycrystalline ice during high-temperature simple shear, The Cryosphere, 13, 1495–1511, https://doi.org/10.5194/tc-13-1495-2019, 2019.

Lilien, D. A., Rathmann, N. M., Hvidberg, C. S., and Dahl-Jensen, D.: Modeling Ice-Crystal Fabric as a Proxy for Ice-Stream Stability, J. Geophys. Res. Earth Surf., 126, e2021JF006306, https://doi.org/10.1029/2021JF006306, 2021.

Qi, C., Prior, D. J., Craw, L., Fan, S., Llorens, M.-G., Griera, A., Negrini, M., Bons, P. D., and Goldsby, D. L.: Crystallographic preferred orientations of ice deformed in direct-shear experiments at low temperatures, The Cryosphere, 13, 351–371, https://doi.org/10.5194/tc-13-351-2019, 2019.

Richards, D. H. M., Pegler, S. S., Piazolo, S., and Harlen, O. G.: The evolution of ice fabrics: A continuum modelling approach validated against laboratory experiments, Earth Planet. Sci. Lett., 556, 116718, https://doi.org/10.1016/j.epsl.2020.116718, 2021.

Thorsteinsson, T., Waddington, E. D., and Fletcher, R. C.: Spatial and temporal scales of anisotropic effects in ice-sheet flow, Ann. Glaciol., 37, 40–48, https://doi.org/10.3189/172756403781815429, 2003.

---

## Referee Comment (RC2)

**Review of Llorens et al.  The Cryosphere - *MS No.: tc-2021-224***

**Can changes in ice-sheet flow be inferred from crystallographic preferred orientations?**

Overall, I find this paper is presenting some oversimplified steady-state models for the evolution of CPOs in ice, and their total applicability to ice-sheet flow has to be questioned. Both thermal conditions, strain cycling and strain localization, particularly within the middle sections of polar ice sheets are far from steady state. This poses a real challenge for interpreting CPOs and is a problem that is not mentioned in this paper. In this contribution the authors are avoiding and fail to point out these issues. Only if significant modifications are undertaken, then this paper will make a suitable contribution for readers of 'The Crysophere'.

The first two parts of this contribution (sections 1-2 & Abstract) presents a very poorly described introduction to a set of numerical models to simulate CPO evolution during ice flow. In these sections there are very significant problems with the construction and poor referencing to previous works, which are relevant to the focus of this paper.  It is quite obvious that none of the 9 other co-authors with a good command of the English language have edited or read the first part of this manuscript.

It would be a benefit to the reader if there was some reorganization of the text. It would be good if sections 3.2.1 to 3.2.4 were taken out of the 'Methods' section and combined with the appropriate sub-sections in 4 'CPO evolution results'. This would provide a clearer pathway into what the modelling is trying to achieve.

I also feel that there is an over citation of papers, which have little value to the main thrust of this manuscript and should be deleted. Or are constantly cited for little reason and just disrupts the text; this is particularly so in the introduction and discussion.

In its present state, portions of this paper need some major rewriting and possible rearrangements. I feel the authors could modify their figures to reduce the magnitude of strains displayed, as these strains are not what is encountered in the majority of natural ice streams because the strain is competing with recrystallization and basal processes.

**Title page & authorship:**

The 'Title page" shows 10 contributing authors. It appears Llorens may have been the sole author for parts of this paper and that this version of the manuscript has not been thoroughly scrutinized by all her co-workers.

In fact, this co-authorship issue should accord with the Vancouver protocol: ([http://www.icmje.org/recommendations/browse/roles-and-responsibilities/defining-the-role-of-authors-and-contributors.html](http://www.icmje.org/recommendations/browse/roles-and-responsibilities/defining-the-role-of-authors-and-contributors.html)). An author must have
- Made substantial contributions to the conception or design of the work; or the acquisition, analysis, or interpretation of data for the work; AND
- Been involved in drafting the work or revising it critically for important intellectual content; AND
- Provided final approval of the version to be published; AND
- Agreed to be accountable for all aspects of the work in ensuring that questions related to the accuracy or integrity of any part of the work are appropriately investigated and resolved.

Hence if these co-workers have not had input into this version of the paper (the AND), then many of these contributors should be listed in the acknowledgements and not cited as authors.

In the Title, some of the wording (e.g., **changes in ice-sheet flow)** seems inappropriate as the manuscript is more about **strain regimes**. Do the authors want to change the title?

**Abstract:**

This is terribly written and has absolutely no appeal for a reader. The authors don't tell us the type of CPOs developed and fail to explain the changes in the CPOs. They say they are looking at "influence of ice deformation history" during the creep of ice. Well there are other changes in the deformation history that are identified by Peternell & Wilson (2016), processes which are not even mentioned in the current paper. The Abstract needs to be restructured.

**Introduction:**

The first sentence is quite irrelevant to this paper and is essentially the same as Llorens et al. (2016). Again, the following sentence does not reflect the real focus of the paper. The following description does not highlight a problem that will be tackled in this manuscript. Instead it is an incredibly poor introduction to the activation of slip systems in ice and relationship to CPOs. Also get rid of all references to sea level changes – they are not relevant to this paper (e.g., Nerem, Golledge). Other references e.g. Katayama & Karato; and non-scientific articles such as Alley & Joughin should be deleted.

Line 74: It is inappropriate to say "Durham et al., 1983 and many others". Here you should properly cite previous experimental studies – "not many others". I should point out Wilson & Peternell (2012) did describe experiments with a pre-existing CPO.

Lines 71-80: This is an extremely poorly written and referenced section on previous ice experiments. The reference to Gao & Jacka should be deleted as this was a poorly designed set of experiments that go to very low strains. There are many more applicable and recent experiments that should have been referenced see Jacka & Li (2000), Wilson et al. (2020) + Fan et. al (2021) and references therein.

Lines 84-100: Again, a poorly referenced section on numerical models applicable to ice that does not elaborate on the different types of models. There are small strain models such as Wilson & Zhang (1994) and references therein, and othe higher strain model studies that may be worth citing.

Line 89-90: Surely, we don't need all the self-referencing to the Llorens papers and the Piazolo et al. 2019 paper as these are more about recrystallization microstructures than CPO development. These references are really not necessary here or elsewhere in the following text.

**2. Flow regime:**

I have an issue with this section as I do not believe this text or Figure 1 have been constrained by a clear description and appropriate references to what happens in a longitudinal section through an ice stream. I would suggest the authors look at the paper of Donoghue and Jacka (2009) and references therein. With a proper literature search the authors will find there are also other papers o longitudinal strain and shear stresses in ice sheets that need to be consulted. Also, Z is normally not vertical.

Line 111: Why not state at the onset what your different examples are?

There is no clear description of the type of CPOs identified in these different regimes. Is it possible to summarise what is found in nature?

Line 136: the references to Yong, LeDoux, Lutz are not needed here as they are cited again on line 253.

Lines 137-139: should be in figure caption.

Figure 1: This is a very much oversimplified diagram, it is probably worth the authors reading Donoghue and Jacka's (2009) description of the changes along a flow line. We all know there are localized zones of high shear strains in any ice stream (e.g. Thwaites et al. (1984) as such can this diagram be made more realistic. The caption lacks any description of what XYZ stand for and what are the broad arrows on the boxes stand for? I don't follow the changes from regime II to III? I can't see a vertical shear plane in regime IV? Why is Z vertical? We normally consider vertical flattening at the upper levels of ice sheets with the ice deforming mainly by compression along the vertical direction.

**3. Methods:**

Lines 140-153: Could this be shortened as there is major overlap with Llorens et al. (2016a)?

Line 170: Up to now there has been no discussion of three slip systems.

Line 195: Remove {brackets} and replace with (0001). This comment applies to figures 2 to 6. I see that in other papers by Llorens et al. that there is also a use of {} for (0001). Crystallographers use these {} brackets where there are multiple axes and not the unique 0001 axis, hence I would suggest using (). The only justification of using {} would be if you are describing a collection of diffraction reflections (001) + (002) + (003).

Table 1; Where does the strain rate come from in caption? There is no discussion of this in the text.

Table 2:  The caption is extremely poor and needs to be expanded. A series of ice cores and examples are provided in table, but there is no documentation or reference as to who have described these examples. Also ($\varepsilon 1=0.92$) is specified but there is no explanation as to what $\varepsilon 1$ is here or in previous text. In fact, the authors should justify why such high strains are used for an average ice stream.

The different modelling scenarios discussed in section 3.2 are well written and are good summaries. However, it would be better if they were linked to the description of the results.

**4. CPO evolution results:**

Why not take sections 3.2.1 to 3.2.4 out of the 'Methods' section and combined with the appropriate sub-sections here? It would improve the readability of the manuscript.

Figures 3, 4, 5: Again, in this figure and caption why is (0001) put in brackets? On Line 277: "y(red)"? It is not clear how the red associated with '*y*'

The results in these sections are well presented and it is obvious that some authors other than Llorens may have input into writing these sections. I then ask myself, who was not involved and is there justification for them being co-authors on this manuscript?

My major concern here is all the models have been run to very high strains which are really quite unrealistic for the majority of ice streams where transported ice (with a pre-existing

CPO) is piggy-backed on basal ice that is undergoing extensive recrystallization and is in a higher stressed environment.

**5. Discussion:**

The introduction to this section is poorly written and is very biased as to which literature the authors quote regarding previous experimental and numerical simulations. There are constant strain rate ice experiments described by Peternell & Wilson (2016; e.g.Fig.3) up to 20% shortening or 0.2 natural strain, which clearly replicate the pattern produced in the PGR diagrams described in this paper. A comparison to such experiments could be made.

The authors need to remove FSE, FSA & ISA from text and figures (e.g. Fig. 7). In fact, it may incorrect to say FSA are finite stretching axes, then why is Z a shortening axis? The reference to Passchier (1990) is not warranted as such axes were described well before him.

Lines 390-405: Can I suggest you also consult Wilson et al. (2020) as there are aspects in this paper and references therein that have been overlooked by the current authors. I feel this whole section on comparison to the CPOs in natural ice needs a lot of rewriting.

Line 405: Delete "plausible scenarios". They are not plausible as you don't consider the processes of strain cycling as described in Peternell & Wilson (2016) and the effect of temperature.

Line 410: Delete "(i.e., long…FSE)" also delete FSE on lines 433, 435, 462, 463 etc

Line 418: Do you need to have all these references?

Line 458:  What is "it" – correct English.

Overall, I find this discussion section is very biased and the authors should have some statements as to the complication observed in nature and why their transitions in CPOs are difficult to establish in natural ice masses.

**Figure 8:**

The caption here needs further expansion – it is not clear what this diagram is really telling us. Why not remove some of the text (lies 440-450) and put this in caption. What does the vertical dashed line represent?

**Figure 9:**

This is an oversimplified conceptual diagram lacking a lot of detail. It should better show where the areas pure shear vs simple shear, indicate any temperature gradient, hence areas of annealing and zones of higher shear stress. The caption is far too brief and there is no clear indication of what figure B stands for, nor is this referred to it in the text?

Overall this discussion section is very disappointing as these results really need to be compare to real scenarios, e.g. Donoghue and Jacka's (2009). In addition, limitations to the application of the models should also be highlighted.

**6. Conclusions:**

Lines 477-479 point 1: "imposed deformation" This is incorrect as deformation conditions include temperature, strain cycling etc. I would delete this whole statement as it won't be a "quick" process and is not a conclusion coming from these models.

Line 485: Surely it is even lower than this value? See my earlier comments.

Line 493, Point 4: This should be deleted as there are many other complications.

**References**

Line 507: Alley & Joughlin should be pp 551-552. However, it is a very generalised article. If you need a reference, then I suggest the instead use of: Alley, R. B.: Flow-law hypotheses for ice-sheet modeling, J.Glaciol., 38, 245–256, 1992.

There are a large number of papers that should be deleted and other papers that the authors may also consider:

Donoghue, S. and Jacka, T.H. (2009) The stress pattern within the Law Dome Summit to Cape Folger Ice Flow line, inferred from measurements of crystal fabrics.
In Hondoh T. ed. *Physics of ice core records II*. Hokkaido
University Press, Sapporo, 125-135.

Fan, S., Prior, D. J., Cross, A. J., Goldsby, D. L., Hager, T. F., Negrini, M., & Qi, C. (2021). Using grain boundary irregularity to quantify dynamic recrystallization in ice. *Acta Materialia*, *209*, 116810. https://doi.org/10.1016/j.actamat.2021.116810

Jacka TH and Li J (2000) Flow rates and crystal orientation fabrics in compression of polycrystalline ice at low temperatures and stresses. In Hondoh T ed. *Physics of ice core records*. Hokkaido University Press, Sapporo, 83–102.

Peternell, M., Wilson, C.J.L., 2016. Effect of strain rate cycling on microstructures and crystallographic preferred orientation during high-temperature creep. *Geology* **44**, 279-282.

Thwaites RJ, Wilson CJL and McCray AP (1984) Relationship between bore hole closure and crystal fabrics in Antarctic ice core from Cape Folger. J. Glaciol., 30(105), 171–179.

Wilson, C.J.L., Peternell, M., Hunter, N.J.R., Luzin, V., 2020. Deformation of polycrystalline D$_2$O ice: Its sensitivity to temperature and strain-rate as an analogue for terrestrial ice. *Earth and Planet. Sci. Lett,* https://doi.org/10.1016/j.epsl.2019

Wilson, C.J.L., Peternell, M.A. (2012). Ice deformed in compression and simple shear: control of temperature and initial fabric. *Journal of Glaciology* 58, 11-22.

Wilson, C.J.L. and Zhang, Y., 1994. Comparison between experiment and computer modelling of plane strain simple shear ice deformation. *J. Glaciol.*, 40 (134), 46-55.

Chris Wilson

08/10/2021

---

## Author Comment (AC1)

On behalf of the author team, I would like to thank the Editor, Nanna Bjørnholt Karlsson, for handling the review process of our manuscript. We thank the reviewers for their suggestions and comments, which have certainly helped to improve the manuscript. We have applied the changes in the manuscript and replied to the questions raised by the reviewers below. Replies to the reviewer are provided in green font and the new or modified text in the manuscript appears in green italic font.

Best regards,

Maria-Gema Llorens

\_\_\_\_\_

**Reviewer #1:**

**Review of "Can changes in ice-sheet flow be inferred from crystallographic preferred orientations?" by Llorens et al.**

This study uses a full-field model to examine the overprinting of ice-crystal fabrics when two strain regimes are experienced. The aim is to understand how/whether the fabric records the ice-flow history or if it is rather representative only of the in-situ deformation regime. Contrary to the title, I do not think it really evaluates "changes in ice-sheet flow," since simulations intentionally mimic idealized steady-state conditions. Nonetheless, this is an interesting topic that has received other attention in the last year, but this study uses a different approach that likely represents the fabric evolution more accurately. The paper is well organized and the writing is clear, with the exception of some unusual notation that created unnecessary confusion for me. The figures are exceptionally well done (Figure 9 in particular could be used to teach good science communication). The manuscript is relevant for *The Cryosphere* and it could be a valuable contribution once some important issues are addressed.

We are grateful for the reviewer's support and for the useful suggestions for improvement.

**Specific comments:**

1. I think a bit of consideration is needed surrounding what it means to "infer changes in icesheet flow," since the present manuscript does not actually address that question. Fundamentally, all simulations here are steady state; they follow idealized particle paths within a steady-state ice sheet, though of course those particle paths transit multiple strain regimes. Since we may be curious whether the fabric matches the in-situ conditions, the paper could simply be retitled to something like "Do crystallographic preferred orientations always represent in-situ conditions?" with the body essentially left as-is.

We have realised that the title and the use of "changes in ice-sheet flow" can be misinterpreted. The paper deals with changes in the flow kinematics (boundary conditions) that a volume of ice experiences through the ice sheet. We do not refer to a change of the flow of the ice sheet itself. Following the first reviewer's advice, we have now modified the article title to "*Can changes in deformation regimes be inferred from crystallographic preferred orientations in polar ice*?"

If the authors are instead set on addressing a question about flow changes (which is probably

more interesting and relevant for a broader audience), then I think more simulations, as well as some explanation of what that would entail, is needed. I would think some kind of change to flow is needed in the simulations to infer a flow change (i.e. not a change along a particle path, but a change to the large scale flow through which the simulated parcel transits). For example, what does the model say about a transition from a dome to a ridge? How long/over what strain would such a change be evident in the CPO? How about formation of an ice stream? Along with those simulations, extensive evaluation like in line 440 would be warranted (i.e. do those changes manifest unambiguously in the CPO? what could we see in the CPO that allows inference of a flow change?).

The reviewer raises a number of interesting questions that could be addressed by our modelling approach. We, however, here indeed deal with the change along a particle path. This is actually still not well known at all, and needs to be addressed before looking at flow changes. Here we show how the CPO changes from one flow regime to another in four scenarios. From the modelling point of view, it actually does not matter if the change is because the ice-sheet flow changes, or because the material flows from one kinematic regime to another through a steady-state ice sheet. In both cases the material experiences a change, and this is what we analyse. Our simulations show that the amount of strain required to fully reset the CPO according to the new regime is not constant but depends on the starting CPO and the new steady-state CPO. It would indeed be of interest to investigate in the future the systematics of the amount of strain needed for a full transition. Even more challenging would be to investigate if it is possible to see if a CPO is not yet fully reset and can thus indicate a change in the flow conditions. We certainly intend to address such questions in the future, but modelling all these scenarios goes well beyond of the scope of the present contribution.

2. The results are hard to believe until dynamic recrystallization (DRX) is given more consideration. The two citations used to justify its exclusion are both modeling studies that in my view are outliers compared to the conventional wisdom on the effect of migration recrystallization on crystal fabric from ice cores (Faria et al., 2014a), experiments (Fan et al., 2021; Qi et al., 2019; Journaux et al., 2019), and modeling (Richards et al., 2021; Faria et al., 2014b). Migration recrystallization is often described as depending on the stress rather than the strain (Duval and Castelnau, 1995), and so may be particularly relevant for V2 and V4 (near the bottom of the ice sheet or in shear margins) where stresses are presumably high. Moreover, even if we were to assume that the effect of recrystallization were relatively small, why does excluding it better represent how CPO responds to a flow change (as implied by the current version of the manuscript)? This concern is intensified because this study shows that, under lattice rotation, development of the new fabric is strongly dependent on the previous fabric, so might a similar sensitivity apply to DRX? This issue is critical; if recrystallization changes the timescale/strain scale over which fabric persists, then a model of lattice rotation alone cannot accurately capture whether flow history can be inferred (or even whether the fabric matches in the in-situ stress and strain). I think this issue is sufficiently important that consideration of different mechanisms of DRX is needed (i.e. rotation and migration recrystallization). The large strains needed to overprint fabric seem to depend on the precise misorientation of the crystallographic axes relative to the new strain, and it seems plausible that even minor effects of rotation recrystallization could alter this misorientation and thus change the results, even if migration recrystallization does not lead to strong CPOs.

According to Fan et al., 2021, under high temperature and high stress conditions, strain weakening in ice is dominated by CPO development, where grain size reduction plays only a minor role.

We have carried out extensive work on simulating dynamic recrystallization processes, including grain boundary migration, recovery and polygonization, in ice and other minerals. As shown in our previous publications (*e.g.*, Llorens et al., 2016a, 2016b, 2017; Steinbach et al., 2016; Gomez-Rivas et al., 2017). These studies demonstrate that dynamic recrystallisation processes have a minor effect on the CPO development (orientation and strength). Therefore, the implementation of DRX in our model would not change the time scale during the development of a preferred crystallographic orientation significantly. However, recrystallization could reduce the time scale of fabric adjustment in case of (1) nucleation of new, strain-free grains, or (2) due grain boundary sliding causing rotation of small grains. In the first case the microstructure could be influenced by the orientation of these new crystals (Thorsteinsson et al., 2003), weakening the CPO. As shown in experiments (Fan et al., 2020), the activation of grain boundary sliding in the small grains also reduces the CPO strength.

To keep the system simple, we did not include spontaneous nucleation or grain boundary sliding in our model. Since, as pointed out by the reviewer, this issue is important, we now discuss it in a new section of "6. *Model limitations*", including a *supplementary figure 4* with the comparison between simulation with and without DRX:

**"6.** Model limitations**

A number of studies that used the VPFFT-ELLE modelling approach (e.g., in Llorens et al., 2016a, 2016b and 2017; Steinbach et al., 2017 and Gomez-Rivas et al., 2017) have shown that dynamic recrystallisation processes, including grain boundary migration, recovery and polygonization, have a minor effect on the CPO development (orientation and strength). The implementation of DRX in our models does not change the time scale during the development of a preferred crystallographic orientation, as shown in supplementary figure 4, where series with and without DRX are compared. However, this may depend on the particular recrystallisation process, for example, (1) spontaneous nucleation (i.e., nucleation of new, strain-free grains), or (2) rotation of small grains due to grain boundary sliding. In the first case the microstructure would be influenced by the orientation of these new crystals (Thorsteinsson et al., 2003), weakening the CPO. As shown in experiments (Fan et al., 2020) activation of grain boundary sliding in the small grains also reduces the CPO strength. It however remains unclear to which extent these two recrystallisation processes play a role in the natural deformation of ice"

**Supplementary figure 4.**

3. I do not think that V3 is an accurate representation of a ridge. Almost by definition, a ridge experiences confined compression/extension rather than pure extension, so the deformation gradient at the ridge itself is

*∇u*=(*a* 0 0

0 –*a* 0

000)

for some a. Of course, some areas can have flow convergence as ice leaves the ridge, in which case we have something like

 $\nabla u = (a+b \ 0 \ 0)$

0 *-a* 0

0 0 -*b*)

but to my knowledge b < a/2 in such areas; the same would be true for ice streams. The  $\nabla u = (a \ 0 \ 0)$

0 *-a* /2 0

0 0 - a/2)

used in the manuscript will have a greater tendency to form a girdle since the extensional stress is equal in all directions in a vertical plane. Because this may affect the results, I would like to see series B, C, and D redone with more realistic conditions, or at least a sensitivity test with

∇*u*=(*a* 0 0 0 −*a* 0

000)

We have tested the B, C and D series, deformed by V'3 in pure shear, as indicated by the referee (*supplementary figures 1, 2 and 3*). The description of the series is now indicated in the text, and the simulation results shown in the supplementary material compared with the current ones. This comment will be included in the text: *As observed in the supplementary figures 1, 2 and 3, not so much influence on the final CPO is observed considering V'3 or V'3.*

---

## Author Comment (AC2)

On behalf of the author team, I would like to thank the Editor, Nanna Bjørnholt Karlsson, for handling the review process of our manuscript. We thank the reviewers for their suggestions and comments, which have certainly helped to improve the manuscript. We have applied the changes in the manuscript and replied to the questions raised by the reviewers below. Replies to the reviewer are provided in green font and the new or modified text in the manuscript appears in green italic font.

Best regards,

Maria-Gema Llorens

**Reviewer #2:**

All the reviewer's suggestions for improvement have been considered and addressed. We have carried out significant modifications based on the issues raised. According to the journal's policy, we have not addressed those comments questioning authorship or asking for native speaker reviewing of the language.

Overall, I find this paper is presenting some oversimplified steady-state models for the evolution of CPOs in ice, and their total applicability to ice-sheet flow has to be questioned. Both thermal conditions, strain cycling and strain localization, particularly within the middle sections of polar ice sheets are far from steady state. This poses a real challenge for interpreting CPOs and is a problem that is not mentioned in this paper. In this contribution the authors are avoiding and fail to point out these issues. Only if significant modifications are undertaken, then this paper will make a suitable contribution for readers of 'The Crysophere'.

The first two parts of this contribution (sections 1-2 & Abstract) presents a very poorly described introduction to a set of numerical models to simulate CPO evolution during ice flow. In these sections there are very significant problems with the construction and poor referencing to previous works, which are relevant to the focus of this paper. It is quite obvious that none of the 9 other co-authors with a good command of the English language have edited or read the first part of this manuscript.

We have included more references on previous works, including the references suggested by the reviewer: Wilson et al., 1994; Wilson et al., 2012; Peternell and Wilson, 2016; Wilson et al., 2020.

It would be a benefit to the reader if there was some reorganization of the text. It would be good if sections 3.2.1 to 3.2.4 were taken out of the 'Methods' section and combined with the appropriate sub-sections in 4 'CPO evolution results'. This would provide a clearer pathway into what the modelling is trying to achieve.

According to this, and a similar suggestion by reviewer #3, qualitative descriptions in 3.2.1 – 3.2.4 appear now at the beginning of the corresponding parts of the results section.

I also feel that there is an over citation of papers, which have little value to the main thrust of this manuscript and should be deleted. Or are constantly cited for little reason and just disrupts the text; this is particularly so in the introduction and discussion.

On the one hand, the reviewer considers that some references in the introduction are overcited, and we should delete references. But on the other hand, this comment contradicts the previous request "*In these sections (introduction) there are poor referencing to previous work*". Now, with all the Wilson references included in the text, we hope that this issue is satisfied.

In its present state, portions of this paper need some major rewriting and possible rearrangements. I feel the authors could modify their figures to reduce the magnitude of strains displayed, as these strains are not what is encountered in the majority of natural ice streams because the strain is competing with recrystallization and basal processes.

As explained before, we found that dynamic recrystallization (DRX) does not have a first-order effect on CPO. We found that the deformation geometry dominates CPO evolution. Strain is a different entity than DRX. DRX cannot compete with strain, but only with the effect of strain on the microstructure or CPO. According to the suggestion by reviewer #1, the manuscript now includes a new section "6. *Model limitations*", where the influence of *DRX* in our model is discussed (see reply to reviewer #1 question)

**Title page & authorship:**

The 'Title page" shows 10 contributing authors. It appears Llorens may have been the sole author for parts of this paper and that this version of the manuscript has not been thoroughly scrutinized by all her co-workers.

This is an offensive comment, and we are not replying to it. This has been discussed with the editor.

In fact, this co-authorship issue should accord with the Vancouver protocol: (http://www.icmje.org/recommendations/browse/roles-and-responsibilities/defining-the-role-of-authors-and-contributors.html). An author must have - Made substantial contributions to the conception or design of the work; or the acquisition, analysis, or interpretation of data for the work; AND - Been involved in drafting the work or revising it critically for important intellectual content; AND - Provided final approval of the version to be published; AND - Agreed to be accountable for all aspects of the work in ensuring that questions related to the accuracy or integrity of any part of the work are appropriately investigated and resolved. Hence if these co-workers have not had input into this version of the paper (the AND), then many of these contributors should be listed in the acknowledgements and not cited as authors.

This is an offensive comment, and we are not replying to it. This has been discussed with the editor.

In the Title, some of the wording (e.g., **changes in ice-sheet flow)** seems inappropriate as the manuscript is more about **strain regimes**. Do the authors want to change the title?

Many thanks for this suggestion. We agree that the title of the manuscript should be changed. We consider that *deformation* is more appropriated than *strain*, because strain is a reference to the symmetric part of the deformation. And in this work, we also perform simulations in simple shear conditions, which implies vorticity or asymmetric part of the tensor.

According to this, we have modified it to *"Can changes in deformation regimes be inferred from crystallographic preferred orientations in polar ice?"*

**Abstract:**

This is terribly written and has absolutely no appeal for a reader. The authors don't tell us the type of CPOs developed and fail to explain the changes in the CPOs. They say they are looking at "influence of ice deformation history" during the creep of ice. Well there are other changes in the deformation history that are identified by Peternell & Wilson (2016), processes which are not even mentioned in the current paper. The Abstract needs to be restructured.

Peternell and Wilson discussed the effect of strain rate variations, or non-steady state deformation, on the microstructure by the observed *DRX* processes. Accordingly, this is not a crucial paper regarding the approach used in our manuscript. We do not feel that the abstract was that terrible or should be totally restructured because of a previous publication by the reviewer, which only marginally relevant to our steady-state conditions and without *DRX*. However, in the revised version of the manuscript we now mention this paper as previous work related to the overall topic exposed here.

**Introduction:**

The first sentence is quite irrelevant to this paper and is essentially the same as Llorens et al. (2016). Again, the following sentence does not reflect the real focus of the paper. The following description does not highlight a problem that will be tackled in this manuscript. Instead it is an incredibly poor introduction to the activation of slip systems in ice and relationship to CPOs. Also get rid of all references to sea level changes – they are not relevant to this paper (e.g., Nerem, Golledge). Other references e.g. Katayama & Karato; and non-scientific articles such as Alley & Joughin should be deleted.

The study of ice rheology has the ultimate aim of understanding ice-sheet flow, and thus its discharge to the ocean and impact on sea level rise. We consider that this aspect has to be pointed out in the paper in the general introduction because it has a direct implication for societal challenges. This is currently just mentioned in two sentences, including two general references (which only represent 9% of the introduction).

On the other hand, the activation of slip systems and CPO's, that the referee thinks that it is poorly explained, takes 39 lines (lines 45-84), which represents the 74% of the introduction.

We agree that Alley and Joughin reference was not appropriate, and we have replaced it to Alley (1998) following the reviewer's recommendation.

Line 74: It is inappropriate to say "Durham et al., 1983 and many others". Here you should properly cite previous experimental studies – "not many others". I should point out Wilson & Peternell (2012) did describe experiments with a pre-existing CPO.

See the reply to the following question.

Lines 71-80: This is an extremely poorly written and referenced section on previous ice experiments. The reference to Gao & Jacka should be deleted as this was a poorly designed

set of experiments that go to very low strains. There are many more applicable and recent experiments that should have been referenced see Jacka & Li (2000), Wilson et al. (2020) + Fan et. al (2021) and references therein.

We have modified the references in this paragraph according to this and the previous suggestion, including all the references by the reviewer: *"Experimental studies have utilised ice to understand how CPOs develop and evolve under deformation (see Kamb et al., 1972; Wilson, 1982; Jacka and Macagnan, 1984, Wilson and Peternell, 2012; Budd et al., 2013; Montagnat et al., 2015; Vaughan et al., 2017, Fan et al., 2020). Most studies by far use bulk isotropic ice (i.e., with a random CPO) that is then subjected to a single deformation event. Due to the limitations of laboratory deformation experiments, to our knowledge only a few studies have used polar ice samples starting with a pre-existing CPO. Moreover, most of such studies focused on vertical uniaxial compression of samples with a pre-existing CPO that was formed by vertical compression (resulting in a c-axis cluster or cone) (Azuma and Higashi, 1984; Dahl-Jensen et al., 1997, Castelnau et al., 1998, Jacka and Li, 2000, Fan et al., 2020).*

Lines 84-100: Again, a poorly referenced section on numerical models applicable to ice that does not elaborate on the different types of models. There are small strain models such as Wilson & Zhang (1994) and references therein, and other higher strain model studies that may be worth citing.

We have included more references of numerical modelling of ice according to this comment. However, we think that we should here cite only references to the state of the art on numerical modelling, so we will not include Wilson and Zhang (1994). The paragraph now includes Montagnat et al. (2011) and Llorens et al. (2017, 2020).

Line 89-90: Surely, we don't need all the self-referencing to the Llorens papers and the Piazolo et al. 2019 paper as these are more about recrystallization microstructures than CPO development. These references are really not necessary here or elsewhere in the following text.

Llorens et al., papers focus both recrystallisation microstructures and CPO development. However, we have moved these references to the previous paragraph (see reply above), but we have kept Montagnat et al. (2014b) and Piazolo et al. (2019) because we consider them two relevant review papers referring to numerical modelling of polycrystalline ice.

**2. Flow regime:**

I have an issue with this section as I do not believe this text or Figure 1 have been constrained by a clear description and appropriate references to what happens in a longitudinal section through an ice stream. I would suggest the authors look at the paper of Donoghue and Jacka (2009) and references therein. With a proper literature search the authors will find there are also other papers o longitudinal strain and shear stresses in ice sheets that need to be consulted. Also, Z is normally not vertical.

Z is now shown as the vertical axis.
See comments below.

Line 111: Why not state at the onset what your different examples are?

Line 111 has been moved to the end of the paragraph, and the description of the considered deformation regimes is now at the beginning of the paragraph.

There is no clear description of the type of CPOs identified in these different regimes. Is it possible to summarise what is found in nature?

We have included in this section the CPO description from natural observation at different regimes in this chapter 2:

- *"Observations from ice core natural samples at these domains, range from a vertical single maximum to a vertical girdle (Montagnat et al., 2012; Weikusat et al., 2017)"*
- *"In depth, observations from ice cores indicate a vertical single maximum."*
- *"Inside a glacier, ice stream or in a flank flow (Voigt, 2017), flow acceleration may dominate, resulting in extension along the flow direction (zone III in Fig. 1a), observed by a vertical girdle in ice core samples (Voigth, 2017)"*

Line 136: the references to Yong, LeDoux, Lutz are not needed here as they are cited again on line 253.

Removed.

Lines 137-139: should be in figure caption.

Moved to figure caption.

Figure 1: This is a very much oversimplified diagram, it is probably worth the authors reading Donoghue and Jacka's (2009) description of the changes along a flow line. We all know there are localized zones of high shear strains in any ice stream (e.g. Thwaites et al. (1984) as such can this diagram be made more realistic. The caption lacks any description of what XYZ stand for and what are the broad arrows on the boxes stand for? I don't follow the changes from regime II to III? I can't see a vertical shear plane in regime IV? Why is Z vertical? We normally consider vertical flattening at the upper levels of ice sheets with the ice deforming mainly by compression along the vertical direction.

In the revised version, the figure caption now includes the description of the xyz coordinates. We have modified the sketch of regime IV to indicate the vertical shear plane. The *z*-coordinate is now the vertical. This is a simplified diagram, as explained in the text. In order to transfer simulations conditions, we think we need these simplifications here. Of course, we agree, that more complex scenarios can occur in nature, e.g. in small ice caps, but this goes beyond the scope of this contribution. However, according to this comment and also from reviewer #3, we have now more clearly stated in the text that the applied velocity gradient fields applied are assumptions.

[Figure]

**3. Methods:**

Lines 140-153: Could this be shortened as there is major overlap with Llorens et al. (2016a)?

*We have shortened this paragraph by removing the following sentence: "ELLE has been successfully used for the simulation of a variety of studies of rock microstructure evolution during deformation and metamorphism (see Piazolo et al., 2019)."*

Line 170: Up to now there has been no discussion of three slip systems.

*We believe this line to be the appropriate place to explain the slip systems considered in our simulations.*

Line 195: Remove {brackets} and replace with (0001). This comment applies to figures 2 to 6. I see that in other papers by Llorens et al. that there is also a use of {} for (0001). Crystallographers use these {} brackets where there are multiple axes and not the unique 0001 axis, hence I would suggest using (). The only justification of using {} would be if you are describing a collection of diffraction reflections (001) + (002) + (003).

*Point taken. We have corrected the notation in the whole manuscript.*

Table 1; Where does the strain rate come from in caption? There is no discussion of this in the text.

*We have removed the strain rate from the figure caption, as now the approximately time to destroy a fabric is calculated and included in the text, assuming the natural strain rates for every ice-sheet domain (see reply to reviewer #1).*

Table 2: The caption is extremely poor and needs to be expanded. A series of ice cores and examples are provided in table, but there is no documentation or reference as to who have described these examples.

*The caption has now been extended including the references to the ice cores: "Table 2. Deformation regimes applied to the different series of numerical simulations, including idealised deformation regimes in drill cores and examples. References of ice core*

*description: GRIP (Thorsteinsson et al., 1997), GISP2 (Gow et al., 1997), Dome C (Durand et al., 2007), Dome F (Azuma et al., 1999), Talos Dome (Montagnat et al., 2012), NorthGrip (Faria et al., 2014); EDML (Weikusat et al., 2017), NEEM (Montagnat et al., 2014a), Vostok (Lipenkov et al., 1998)"*

Also ($\varepsilon 1=0.92$) is specified but there is no explanation as to what $\varepsilon 1$ is here or in previous text. In fact, the authors should justify why such high strains are used for an average ice stream.

In our simulations, at $\varepsilon_1=0.92$, the microstructure develops an approximately 80% of end-member CPO (P or G). We consider that it fits with reaching the "secondary creep" quasi steady state in deformation tests at ca. 1% (Bud & Jacka, Treverrow).

The different modelling scenarios discussed in section 3.2 are well written and are good summaries. However, it would be better if they were linked to the description of the results.

According to this, and a similar suggestion by reviewer #3, we have merged sections 3 and 4. Qualitative descriptions in 3.2.1 – 3.2.4 appear now at the beginning of the corresponding parts of the results section.

**4. CPO evolution results:**

Why not take sections 3.2.1 to 3.2.4 out of the 'Methods' section and combined with the appropriate sub-sections here? It would improve the readability of the manuscript.

According to this comment, and a similar comment by reviewer #3, we have extended the description of the methods in section 3 and moved sections from 3.2.1 to 3.2.4 to section 4. Results. See reply above.

Figures 3, 4, 5: Again, in this figure and caption why is (0001) put in brackets? On Line 277: "y(red)"? It is not clear how the red associated with '*y*'

Brackets are corrected in the whole manuscript (see reply above). It is a mistake; it should indicate *black* instead of *red* in the text.

The results in these sections are well presented and it is obvious that some authors other than Llorens may have input into writing these sections. I then ask myself, who was not involved and is there justification for them being co-authors on this manuscript?

This is a very inappropriate comment, and we are not replying to it following the editor's recommendation.

My major concern here is all the models have been run to very high strains which are really quite unrealistic for the majority of ice streams where transported ice (with a pre-existing CPO) is piggy-backed on basal ice that is undergoing extensive recrystallization and is in a higher stressed environment.

This issue was already addressed in our answer to reviewer #1. Clearly, when a volume of ice is piggy-backed, it does not experience a change of flow regime, or at least not one that reaches any significant strain.

**5. Discussion:**

The introduction to this section is poorly written and is very biased as to which literature the authors quote regarding previous experimental and numerical simulations. There are constant strain rate ice experiments described by Peternell & Wilson (2016; e.g.Fig.3) up to 20% shortening or 0.2 natural strain, which clearly replicate the pattern produced in the PGR diagrams described in this paper. A comparison to such experiments could be made.

We have included the reference to Peternell and Wilson in the discussion between experiments and simulation results.

The authors need to remove FSE, FSA & ISA from text and figures (e.g. Fig. 7). In fact, it may incorrect to say FSA are finite stretching axes, then why is Z a shortening axis? The reference to Passchier (1990) is not warranted as such axes were described well before him.

We do agree that these abbreviations may appear confusing to readers that are not used to this terminology. Therefore, these are now explicitly written throughout the text. It is customary in geology to denote the finite stretching axes, which are the principal axes of the finite strain ellipsoid by *X*, *Y*, and *Z*, from longest to shortest. The *Z*-axes (capital) is this the direction of maximum finite shortening, as is explained in the text. The *Z*-axis is not the same as the z-axis (lower case) of the coordinate system. By writing out the terms in full, we think there should be no confusion.

Lines 390-405: Can I suggest you also consult Wilson et al. (2020) as there are aspects in this paper and references therein that have been overlooked by the current authors. I feel this whole section on comparison to the CPOs in natural ice needs a lot of rewriting.

Our simulations do not include DRX, as explained in several replies above (it is now explained in the manuscript in chapter 6. Model limitations). Wilson et al., 2020 address the influence of temperature and strain-rate (i.e. DRX) on ice deformation. However, we have included this reference in the text as a general reference for ice deformation experiments.

Line 405: Delete "plausible scenarios". They are not plausible as you don't consider the processes of strain cycling as described in Peternell & Wilson (2016) and the effect of temperature.

We are not considering DRX or variations in strain rate, as Peternell and Wilson did. Addressing this issue, we have included a new section *6. Model limitations* (see reply to reviewer #1).

Line 410: Delete "(i.e., long…FSE)" also delete FSE on lines 433, 435, 462, 463 etc

See reply above.

Line 418: Do you need to have all these references?

Yes, we do.

Line 458: What is "it" – correct English.

There is no "it" in line 458.

Overall, I find this discussion section is very biased and the authors should have some statements as to the complication observed in nature and why their transitions in CPOs are difficult to establish in natural ice masses.

The reviewer acknowledged in his comments that volumes of ice may experience changes in their flow regime, as investigated in this paper, and by others. The reviewer did not provide any arguments why transitions in CPO are difficult to establish in nature. We see no reason to address this comment.

**Figure 8:**

The caption here needs further expansion – it is not clear what this diagram is really telling us. Why not remove some of the text (lies 440-450) and put this in caption. What does the vertical dashed line represent?

The figure caption has been expanded, including the explanation of the vertical dashed line: *"Evolution of the relative activities of basal, pyramidal and prismatic slip systems during deformation for all series presented, calculated from Equation (1). Transition of deformation regimes are marked with a vertical dashed line. In the A, B and D series, the second flow regime produces a prominent increase in basal slip, while the pyramidal slip system activity is reduced. However, in series C the basal activity remains constant, and the prismatic slip is increased"*

**Figure 9:**

This is an oversimplified conceptual diagram lacking a lot of detail. It should better show where the areas pure shear vs simple shear, indicate any temperature gradient, hence areas of annealing and zones of higher shear stress. The caption is far too brief and there is no clear indication of what figure B stands for, nor is this referred to it in the text?

It is irrelevant indicate the temperature gradient, as we are not considering temperature variations in our approach (*DRX* processes are not included in the models).

On the other hand, we disagree with the suggestion to modify this figure, because the reviewer finds it as an *oversimplification*. In agreement with reviewer #1 we consider that this figure does not need modifications. As reviewer #1 correctly interpreted, the purpose of this figure is to summarise the results of this work for a broad audience in a simple and understandable way. Reviewer #1 says: ***The figures are exceptionally well done (Figure 9 in particular could be used to teach good science communicatio***

Overall this discussion section is very disappointing as these results really need to be compare to real scenarios, e.g. Donoghue and Jacka's (2009). In addition, limitations to the application of the models should also be highlighted.

According to this suggestion, and a similar suggestion by reviewer #1, we have included the new section *"6. Model limitations"* (see reply to reviewer #1).

**6. Conclusions:**

Lines 477-479 point 1: "imposed deformation" This is incorrect as deformation conditions include temperature, strain cycling etc. I would delete this whole statement as it won't be a "quick" process and is not a conclusion coming from these models.

We have changed "imposed deformation" to "imposed boundary conditions".

Line 485: Surely it is even lower than this value? See my earlier comments.

This statement is based on the results of our study. We therefore do not think we need to modify the text here.

Line 493, Point 4: This should be deleted as there are many other complications.

We find the suggestion by reviewer #3 more correct than this one by reviewer #2. We have followed the suggestion by reviewer #3, modifying this line in order to provide an answer to the question expressed in the title of the manuscript "*Can changes in deformation regimes be inferred from crystallographic preferred orientations in polar ice*?". The line has been modified to: *"According to our results, CPOs are reliable indicators of the current flow conditions, as they usually adapt to them in a relatively short time. However, caution is warranted when a volume of ice may have experienced complex (multi-stage) deformation histories."*

References:

Fan, S., Prior, D. J., Cross, A. J., Goldsby, D. L., Hager, T. F., Negrini, M., & Qi, C. (2021). Using grain boundary irregularity to quantify dynamic recrystallization in ice. *Acta Materialia*, *209*, 116810. https://doi.org/10.1016/j.actamat.2021.116810

Jacka TH and Li J (2000) Flow rates and crystal orientation fabrics in compression of polycrystalline ice at low temperatures and stresses. In Hondoh T ed. *Physics of ice core records*. Hokkaido University Press, Sapporo, 83–102.

Peternell, M., Wilson, C.J.L., 2016. Effect of strain rate cycling on microstructures and crystallographic preferred orientation during high-temperature creep. *Geology* **44**, 279-282.

Thwaites RJ, Wilson CJL and McCray AP (1984) Relationship between bore hole closure and crystal fabrics in Antarctic ice core from Cape Folger. J. Glaciol., 30(105), 171–179.

Wilson, C.J.L., Peternell, M., Hunter, N.J.R., Luzin, V., 2020. Deformation of polycrystalline D2O ice: Its sensitivity to temperature and strain-rate as an analogue for terrestrial ice. *Earth and Planet. Sci. Lett,* https://doi.org/10.1016/j.epsl.2019

Wilson, C.J.L., Peternell, M.A. (2012). Ice deformed in compression and simple shear: control of temperature and initial fabric. *Journal of Glaciology* 58, 11-22.

Wilson, C.J.L. and Zhang, Y., 1994. Comparison between experiment and computer modelling of plane strain simple shear ice deformation. *J. Glaciol.*, 40 (134), 46-55.

---

## Author Comment (AC3)

On behalf of the author team, I would like to thank the Editor, Nanna Bjørnholt Karlsson, for handling the review process of our manuscript. We thank the reviewers for their suggestions and comments, which have certainly helped to improve the manuscript. We have applied the changes in the manuscript and replied to the questions raised by the reviewers below. Replies to the reviewer are provided in green font and the new or modified text in the manuscript appears in green italic font.

Best regards,

Maria-Gema Llorens
* * *
**Reviewer #3**

The paper provides a systematic modeling study to examine the effect of pre-existing CPO on final CPO in scenarios where stress state changes in the deformation history. The 4 simplified scenarios are well described and the results are interesting and can help inform interpretation of past flow inferred from core sample microstructures. I recommend publication after some modification.

We greatly appreciate the reviewer's support for publication, and their suggestions for improvement.

My main issue is that it feels a little like a black box. I don't understand how you go from the physics of deformation described in section 2 to the results in, say figure 3c. I don't think you need to provide post-processing and exhaustive details, but it seems like it would help readers like myself who do not model if there was a very simple description of how you get figure 3c, so that a person doesn't have to go to Llorens et al. 2016 for the background needed.

We have extended the postprocessing explanation for the PGR diagram (figure 3c) in the manuscript with to this comment: *"Crystal symmetry shows the relative proportion of point (P), girdle (G) and random (R) components of the (0001) crystallographic axis, or c-axis distribution in a ternary diagram (Vollmer, 1990). The P, G and R proportion is calculated from the three eigenvalues ($a_1$, $a_2$, $a_3$) as $P=a_1 - a_2$, $G=2a_2 - a_3$ and $R=3a_3$."*

*In order to explain how figures are obtained, we include an additional figure where the workflow can be visualised (supplementary figure 5)*

[Figure]

*supplementary figure 5*

Additionally, I provide line by line comments that may help improve readability.

Minor comments:

Line 36: Recommend deleting the word "on". Deleted.

Line 49: consider reordering this sentence to be clearer. Maybe more like: "Polycrystalline ice (ice Ih) in glaciers, ice sheets, and ice shelvesflows in response to gravitational forces." Point taken. Changed to: "*Polycrystalline ice (ice 1h) in glaciers, ice sheets and ice shelves flows in response to gravitational forces (e.g., Hudleston, 2015*)".

Line 73: The Durham et al., 1983 paper doesn't have anything to do with CPO development or evolution so isn't appropriate there. Perhaps something by Montagnat?

According to this comment, and also a similar comment by reviewer #2, we have removed the Durham reference and now include the following references: *"Experimental studies have utilised ice to understand how CPOs develop and evolve under deformation (see Kamb et al., 1972; Wilson, 1982; Jacka and Macagnan, 1984, Wilson and Peternell, 2012; Budd et al., 2013; Montagnat et al., 2015; Vaughan et al., 2017, Fan et al., 2020)".*

Line 80: I would add Fan et al. 2020 to this list. I do see you mention it later in the paper, but would be good here as well.

This sentence refers to experiments starting with a pre-existing CPO, not the case of Fan et al. (2020). However, we have added this reference in the previous paragraph, where we find it fits better (see reply above).

Line 92: consider providing additional refs here to put this work into context with previous modeling efforts

According to this comment, and a similar comment by reviewer #2, we have modified the whole paragraph including references to recent numerical studies: *"Moreover, numerical*

*simulations of polycrystalline ice and their comparison with experimental and natural data provide useful insights into CPO development, as they allow visualizing and quantifying the microstructural evolution up to high strain (Montagnat et al., 2014b; Piazolo et al., 2019).). However, as in the case of laboratory experiments, most numerical studies to date have focused on systems that start with an initially random CPO to which a single deformation event is applied (Montagnat et al., 2011; Llorens et al., 2017,2020)."*

Line 98: sorry, I don't know what a cloudy band is in this context…perhaps define? Are they layers containing dust particles? Perhaps explain why this is or isn't relevant to the effort here

For sake of simplicity, we have modified the paragraph for simplification: *"Jansen et al. (2016), where the viscoplastic response of ice polycrystals with a starting CPO is studied"*

Line 100: perhaps another half sentence for the non-glaciologists: "…experiences multiple changes in deformation regime during ice-sheet flow as it _____" (I don't know, changes course and rounds topographical features?…just a flavor of the type of changes made for those who don't know)

The sentence has been changed to: *"Considering that polar ice typically experiences multiple changes in the deformation regime during ice-sheet flow, such as the transition from the co-axial strain in the centre of the ice mass to non-coaxial strain at depth and away from the centre (Jennings and Hambrey, 2021), systematic studies providing a comprehensive understanding of CPO development during multi-stage deformation histories are essential."*

Line 107, 369: in intro you didn't use an apostrophe in CPOs for plural. I don't know which it should be, but just be consistent

Point taken. Corrected to CPOs in the whole manuscript.

Line 110: perhaps a sentence here to say something along the lines of flow in nature is complicated, but for ease of understanding you provide the simplified diagram in Figure 1, which divides the flow patterns into four distinct zones. If you are ignoring some aspects of flow (T?) then describe here.

We have included the following sentence at the beginning of chapter 2: *"We analyse different examples of flow changes that represent relevant and/or common deformation regimes in ice sheets, assuming a constant strain rate and temperature."*

Line 163: I recommend deleting "an"

Corrected.

Line 169: here you define n as the rate sensitivity exponent, but all other occurrences you call it the stress exponent. If you mean the same thing, I recommend calling it the stress exponent here.

Modified now to stress exponent.

Line 205 to 260: I recommend more clearly stating how you came up with the velocity gradient tensors for each zone. It is not clear if this should be a result or an assumption. If it is

an assumption, I recommend more clearly stating that and have this section just be stating that you will run 4 series that represent different transitions from one V to another V, basically introducing Table 2. I would save the qualitative descriptions currently in 3.2.1 – 3.2.4 to instead appear at the beginning of results for each of those series.

*We have more clearly stated that the gradient tensors for each zone are assumptions, and we have introduced table 2 in the text: "We considered four different model series (from series A to D) to simulate flow transitions between pairs of deformation regimes (V that dominate in different zones of the ice mass through which a volume of ice may travel (Figure 1).*

*Series A represents ice flowing from the centre of the dome to deep lateral zones (from zone I to zone II in Fig. 1). To simulate this transition, we carried out a series of simulations with first vertical uniaxial compression parallel to y ($V_1$), followed by dextral simple shear in the vertical plane (xz) ($V_2$) (Table 1 and 2). Similar to A, Series B shows the transition of ice flowing centre parts of the ice sheet, but in this case from the centre of the ridge to deep lateral zones (from zone I to zone II in Fig. 1). For series B, we considered that the ice aggregate is first deformed by $V_3$, horizontal uniaxial extension parallel to x, followed by $V_2$, dextral simple shear in the vertical plane (xz) (Table 1 and 2). Series C simulates ice flowing from an ice dome to an ice flank or stream (from zone I to zone III in Figure 1). Series C was carried out assuming first vertical uniaxial compression parallel to z ($V_1$) followed by uniaxial extension parallel to x ($V_3$) (Table 1 and 2). Finally, series C represents ice flowing from an ice-stream or glacier to an ice shelf or shear margin (from zone III to zone IV in Figure 1). For this series, we considered first uniaxial extension in the x direction ($V_3$), and subsequently dextral simple shear in the horizontal plane (xy) ($V_4$) (Table 1 and 2). For comparison, simulations of microstructures deformed under single-deformation event ($V_2$, $V_3$ and $V_4$) are shown together will all series results."*

According to a similar suggestion from reviewer #2, we have merged sections 3 and 4. The descriptions in 3.2.1 – 3.2.4 now appear at the beginning of the corresponding parts of the results section.

Line 209: recommend changing "examples" to "example"

Corrected.

Line 223: recommend making "simulation" plural

Corrected.

Line 256: recommend deleting "of" and "before" from this sentence.

Corrected.

Line 374 (but really 366 – 381): It is unclear where in this paragraph you are referring to historically, as in past studies, and where you mean the results from this study. Try to make it very clear and emphasize how your results confirm or deny previous works by including some words at the beginning of sentences like: "Indeed, our experiments confirm that…." In case the reader does not have prior knowledge of CPO evolution, drag us along very explicitly. [ah, it is much clearer in the 2nd paragraph]

Point taken. We have merged the first and second paragraph in order to avoid confusion about what are results and what are past studies.

Line 429: change "loose" to "lose"

Corrected.

Line 431: recommend changing "effectivity" to "effectiveness"

Corrected.

Line 454: the double negative makes this sentence hard to follow. Consider changing "not destroyed" to "retained"

Addressing this commen,t and those by reviewer #1, the sentence is now modified to: *"Our results suggest that, under natural conditions, as for example those at the onset of the NEGIS onset where the velocity increases by 40 m/yr over a distance of 120 km (i.e. strain rate of ~1x10$^{-11}$ s$^{-1}$; Joughin et al., 2018), an inherited fabric would be preserved for at most for ~ 7 kyr".*

Line 459: if these results are also in agreement (or even if they are not in agreement) with other polycrystalline materials, here would be a good place to mention that. One study that comes to mind is Boneh and Skemer, EPSL 406, 2014, which experimentally looked at this very thing in olivine. Putting your ice modeling results into broader context might be a good idea.

We have included a new paragraph discussing this work and comparing it with similar studies in olivine: *"The entire change of a previous CPO also takes place in other rocks, such as olivine-rich rocks in the upper mantle, where a new CPO quickly develops according to the new imposed boundary conditions. The observed CPO will thus not record the full history of changes in the kinematics of deformation (Kaminski,2004). However, as our results reveal, the re-orientation of an inherited CPO depends both on its intensity and on the orientation with respect to the new stress field. These results are in agreement with observations from olivine experiments, where the pre-existing texture orientation determines the way the texture evolves (Boneh and Skemer, 2014). Accordingly, the deformation history could have an impact on the CPO in areas with complex flow, as in subduction zones (Di Leo et al., 2014; Li et al., 2014)."*

Line 492: italicize c in c-axis.

Corrected in the whole manuscript.

Line 493: perhaps reword number 4 to exactly answer the title of the paper? (even if with a caveat)

Point #4 has been reworded as: *"According to our results, CPOs are reliable indicators of the current flow conditions, as they usually adapt to them in a relatively short time. However, caution is warranted when a volume of ice may have experienced complex (multi-stage) deformation histories."*

*References:*

Boneh, Y. and Skemer, P., The effect of deformation history on the evolution of olivine CPO. *Earth and Planetary Science Letters*, *406*, pp.213-222. 2014

Di Leo, J.F., Walker, A.M., Li, Z.H., Wookey, J., Ribe, N.M., Kendall, J.M. and Tommasi, A., Development of texture and seismic anisotropy during the onset of subduction. *Geochemistry, Geophysics, Geosystems*, *15*(1), pp.192-212. 2014

Jennings, S.J. and Hambrey, M.J., Structures and Deformation in Glaciers and Ice Sheets. *Reviews of Geophysics*, *59*(3), p.e2021RG000743. 2021

Kaminski, E., Ribe, N.M. and Browaeys, J.T., D-Rex, a program for calculation of seismic anisotropy due to crystal lattice preferred orientation in the convective upper mantle. *Geophysical Journal International*, *158*(2), pp.744-752. 2004

Li, Z.H., Di Leo, J.F. and Ribe, N.M., Subduction-induced mantle flow, finite strain, and seismic anisotropy: Numerical modeling. *Journal of Geophysical Research: Solid Earth*, *119*(6), pp.5052-5076. 2014

---

## Author Response (AR2)

On behalf of the author team, I would like to thank the Editor, Nanna Bjørnholt Karlsson, for handling this manuscript. We thank the reviewers for their valuable suggestions and comments, which have certainly helped to improve the manuscript. We have applied the changes in the manuscript and replied to the questions raised by the reviewers below. Our replies to each reviewer question with the action taken on the manuscript are provided in green font.

Best regards,

Maria-Gema Llorens
* * *
REFEREE#1
I am fully satisfied with the extent of revisions that authors have undertaken for this revised version. The revised manuscript and figures are really nice and this will be a good contribution to the field. I especially commend authors for professionally and calmly addressing the constructive aspects of Reviewer #2's review while ignoring the pompous bluster that was wholly inappropriate and unprofessional.
One tiny edit to look out for (in the proof only, I do not require a revision) is at line 552, I would change "a most of" to "at most". No other comments.

Many thanks! This sentence is in line 452. We have corrected it.

REFEREE#2
I maintain that ELLE/VPFFT is an outlier on the importance of recrystallization—and while I do not think it mitigates the importance of the work I find that section 6 needs expansion. In my view, it is simply not good enough to disregard recrystallization affecting CPO based on modeling, which is more-or-less what was done in the review response. The authors also disregard experimental evidence saying that Fan et al. show that "strain weakening in ice is dominated by CPO development, where grain size reduction plays only a minor role" but the question is not about strain weakening. This statement does does not mean that DRX is unimportant for CPO development, and I think Fan et al. demonstrate that (I am aware that there is author overlap with this paper, but I feel the need to point this out nonetheless). For example, the first two rows of their Figure 9 are classic migration recrystallization cones. Qi et al, 2019, show additional fabrics that almost certainly require DRX, as they argue in section 4.6. Indeed, in that paper modeling with ELLE/VPFFT (again, done by/with authors on the present work) does show notable differences with and without DRX in their Figure 9. Given that experimental evidence, as well as ice-core evidence from shear margins that shows fabrics clearly caused by DRX (e.g., Jackson and Kamb, 1997; Gerbi et al., 2021), and the simple fact that even relatively cold ice is over 95% its melting point, I think it is clear that DRX would matter for CPO in some of the cases considered here—if the authors disagree, at a minimum I hope they would concede that it is not well known, and put a little bit more into Section 6. In particular, some citations to acknowledge that there is in fact evidence of recrystallization in natural ice (e.g. citations above, in addition to others that can be found in the Faria reviews), would help the last sentence. Additionally, there should be some consideration of how inaccurately modeled DRX could affect the results (by which I

mean at least nucleation and GBS, processes that even the authors concede could affect the fabric); I appreciate the supplementary figure showing us how the model does with DRX, but want a little more humility on the possibility that the model is imperfect. Again, I am not suggesting more modeling, just a little more consideration of the possibility that DRX may, contrary to what ELLE/VPFFT says, control CPO development in some of the cases considered here. Particularly, this seems likely in shear margins, where there is direct evidence of recrystallization-controlled fabrics. It also seems like the authors should acknowledge that the very slow rates of fabric development by lattice rotation in series C could cause DRX to be of relatively greater importance.

We agree that the role of *DRX* on CPO is a matter of debate, and maybe that of GBS even more. We have extended accordingly the section 6, where different key questions about the effect of DRX in nature are specified and discussed. Moreover, we discuss the effect of these processes in our approach, as shown in in suppl. figure 4, where adding GBM, recovery and polygonisation in our modelling scheme has virtually no effect on the CPO. However, these processes do result in a significantly different microstructure (grain shape, size, etc.), as was shown in previous papers by Llorens et al. and Steinbach et al. Therefore, there is no need to add these processes to the current simulations, as we are only concerned with the CPOs. As discussed in the revised section 6, we do realise that this does not necessarily mean that in nature DRX has no effect on CPO. At the moment, we unfortunately cannot include GBS in our simulation code. This is a limitation that we now also clearly acknowledge. We have included the following text and changed the title of the section including "*further processes*":

*6. Model limitations and further processes*

[revised manuscript text omitted]

Specific comments:
L50: A fabric reference is inappropriate for such a basic claim about ice dynamics. Perhaps Cuffey and Paterson is most fitting, or Aggasiz, Forbes, or Tyndall from the 1800s?
We have modified the reference to Cuffey and Paterson.
Fig 5. A label on the green, downward arrow would be helpful
We have included "single regime" on the green arrow.
L451: Typo muddles the meaning; unclear if "a most of" means an upper limit or a best approximation.
According to reviewer #1 we have modify it to "at most".
L481: What does effectivity mean here? Effectiveness at what? Perhaps this is jargon which which I am not familiar, but I suggest a different word choice for clarity.
We have changed effectivity to "The influence of the second flow regime on the reorientation of the inherited CPO".
L494: It would seem that this is contradicted by series C.

Included "with the exception of series C, when a strong point maximum CPO developed during the first deformation regime".

L497: Where along the margins—they vary substantially.

We consider that the paragraph is clear enough now and it already includes this information.

L566: I think conclusion 4 should be deleted, since the two sentences essentially contradict each other. If it were a reliable indicator, then we would not need caution (we know present-day deformation better than past, so trying to infer flow from CPO would happen most often in areas with multi-stage history).

We agree. We have modified the conclusion 4 to "According to our results, CPOs are reliable indicators of the current flow conditions, as they usually adapt to them in a relatively short time. However, caution is warranted when a volume of ice may have experienced consecutive flow events with the extension direction in the same direction.


EDITOR

Fig. 1: For consistency and readability, please move "I" and "II" outside of the ice so they are placed above the ice sheet in the same way as "III" and "IV"

Done.

Line 152: The sentence "At depth, observations from ice cores indicate a vertical single maximum" is unclear. Does this refer to ice at a ridge? Or away from a ridge? Also, which observations? There should be a reference here.

Modified to "At depth, microstructural descriptions from ice cores performed in domes and ridges indicate a vertical single maximum (Thorteinsson et al., 1997; Azuma et al., 1999; Durand et al., 2007; Faria et al., 2014; Weikusat et al., 2017)"

Line 158: What is "zone IVa"? I don't see a IVa in the figure.

Corrected to zone IV

Line 217: "coherent" -> "consistent"?

"coherent" is in line 192. We have modified it to "consistent"

p. 10 (and elsewhere): "Table 1 and 2" should be "Tables 1 and 2"

Corrected.

Line 400: Figs. 3 and 4 do not show the final CPO after simple shear only. Do you mean Fig. 5a?

We have modified the text to "Although the final CPO symmetry is coherent with simple shear deformation, its shape after a strain of $\varepsilon2=4$ still differs from that of the previous case (series A) (see the last step for the second regime in Figs. 3a and 4a) or that of simple shear only (see figure 5 in Llorens et al., 2017).

Line 440: "... the final CPO continues being dominated..." -> "... the final CPO continues to be dominated..."

Corrected.

Line 552: "2,8 kyr" -> "2.8 kyr"

Corrected.
Line 565: "2,5 kyr" -> "2.5 kyr"
Corrected.
Line 603: Flow is not necessarily considerably faster in the NEGIS margins. I suggest rephrasing to clarify that strain is considerably higher in the margins which is really the point.
Modified to "where strain is considerably higher (i.e. strain rate of $\sim$ 4x10-10 s-1)
Fig. 9: The new and the old version are identical?
Yes, according to referee's comments we didn't change this figure.